# Compositional PAC-Bayes:
# Generalization of GNNs with persistence and beyond

**Kirill Brilliantov**
ETH Zürich
kbrilliantov@ethz.ch

**Amauri H. Souza**
Federal Institute of Ceará
amauriholanda@ifce.edu.br

**Vikas Garg**
YaiYai Ltd & Aalto University
vgarg@csail.mit.edu

## Abstract

Heterogeneity, e.g., due to different types of layers or multiple sub-models, poses key challenges in analyzing the generalization behavior of several modern architectures. For instance, descriptors based on Persistent Homology (PH) are being increasingly integrated into Graph Neural Networks (GNNs) to augment them with rich topological features; however, the generalization of such PH schemes remains unexplored. We introduce a novel *compositional* PAC-Bayes framework that provides a general recipe to analyze a broad spectrum of models including those with heterogeneous layers. Specifically, we provide the first data-dependent generalization bounds for a widely adopted PH vectorization scheme (that subsumes persistence landscapes, images, and silhouettes) as well as PH-augmented GNNs. Using our framework, we also obtain bounds for GNNs and neural nets with ease. Our bounds also inform the design of novel regularizers. Empirical evaluations on several standard real-world datasets demonstrate that our theoretical bounds highly correlate with empirical generalization performance, leading to improved classifier design via our regularizers. Overall, this work bridges a crucial gap in the theoretical understanding of PH methods and general heterogeneous models, paving the way for the design of better models for (graph) representation learning. Our code is available at https://github.com/Aalto-QuML/Compositional-PAC-Bayes.

## 1 Introduction

Topological data analysis (TDA) harnesses tools from algebraic topology to unveil the underlying shape and structure of data. TDA has recently gained significant traction within machine learning mainly due to its flagship method: persistent homology (PH) [11], which allows for capturing topological invariants (like connected components and loops) of the input domain at multiple scales. In particular, PH has recently been leveraged as a tool to augment the representational capabilities of graph neural networks (GNNs) [20, 53, 58], with expressivity gains formally established [22, 44]. Intuitively, PH furnishes global structural signatures that complement the local nature of GNNs [5, 18, 20, 57].

Understanding the generalization behavior of these models is crucial as it plays a pivotal role in ensuring their reliability and applicability [41]. In this context, there are two fundamental approaches to achieving generalization bounds: data-independent and data-dependent [47], each offering unique insights into the generalization problem. Both these approaches have been investigated to analyze the generalization ability of GNNs [12, 14, 25, 33, 36, 46, 48, 52, 59]. Data-dependent generalization bounds evoke particular interest since they are typically much tighter than the agnostic bounds afforded by, e.g., VC dimension. However, no such bounds have been unearthed for PH methods (i.e., learnable vectorization schemes) and, consequently, for GNNs enhanced with PH-based descriptors.

We bridge this gap with the first data-dependent generalization bound for classifiers based on a versatile and widely used vectorization framework for persistence diagrams, namely, *PersLay* [5].

38th Conference on Neural Information Processing Systems (NeurIPS 2024).

PersLay leverages *extended persistence* to effectively represent detailed topological features, and subsumes commonly used methods such as persistence landscapes [4], images [1], and silhouettes [6].

Central to our analysis is a novel PAC-Bayes framework (Lemma 2) that provides a general recipe to analyze the generalization of a broad spectrum of models, including those with heterogeneous layers and those comprising multiple sub-models. To achieve this, we introduce general conditions (Equations 6-9) that are satisfied by commonly used learning architectures and, surprisingly, their compositions (Section 4). Leveraging Lemma 2, we show how to obtain bounds for heterogeneous MLPs and GNNs in a straightforward manner (Table 2). Notably, we also establish the first generalization bounds for GNNs augmented with persistence layers (PersLay).

Our exposition focuses on graphs; however, i) our Lemma 2 can be used in any domain and ii) our bound for PersLay considers persistence diagrams obtained from any non-learnable filtration function, and therefore extends more generally to input domains beyond graphs. From a technical perspective, our approach hinges on contrasting previous analyses within the PAC-Bayes framework [9, 37, 38] to extract the common structure encoded in the general conditions of Lemma 2. This allows us to overcome challenges arising from the heterogeneity of the models we consider.

Table 1: Main theoretical contributions of this work.

| **Section 3: Generalized PAC-Bayes** | |
| --- | --- |
| General recipe for heterogeneous models | Lm. 2 |
| Applying the recipe to GNNs and MLPs | Tab. 2 |
| New bound for PersLay | Cor. 1 |
| **Section 4: Compositional PAC-Bayes** | |
| Bound for the composition with MLP | Lm. 3 |
| Bound for two models in parallel | Lm. 4 |
| New bound for GNNs with persistence | Cor. 2 |

Our experiments on several standard real-world datasets confirm strong correlation between the empirical performance and our theoretical bounds. We reinforce the merits of our analysis via regularized PH-based models informed by our bounds with demonstrable empirical benefits.

**Our main contributions** are:

(Theoretical, see Table 1) We develop a general recipe for obtaining PAC-Bayes bounds for a broad class of (possibly heterogeneous) models and their compositions. We also provide the first data-dependent bounds for PH-based classifiers and *combinations* of GNNs and PersLay;

(Empirical) We show that the dependence on parameters depicted in our bounds strongly correlates with the observed performance. We also show that novel regularization schemes based on our bounds can reduce the generalization gap of PH-augmented GNNs on multiple datasets.

## 2 Preliminaries

This section overviews GNNs, persistent homology on graphs, and their combination. We also provide basic notions and results in PAC-Bayes learning, which serve as a background for this work.

**Notation.** We consider attributed graphs denoted as a tuple $G = (V, E, z)$, where $V = \{1, 2, ..., n\}$ is the vertex set, $E \subseteq V \times V$ is the edge set, and $z : V \to \mathbb{R}^{d_z}$ assigns to each vertex $v \in V$ an attribute (or color) $z(v)$. For convenience, hereafter, we denote the feature vector of $v$ by $z_v$. We consider classification tasks with input and label spaces $\mathcal{X}$ and $\mathcal{Y} = \{1, \ldots, K\}$ ($K$ is the number of classes) and the $\gamma$-margin loss $l_\gamma : \mathcal{Y} \times \mathbb{R}^K \to \{0, 1\}$ where $l_\gamma(y, \hat{y}) = \mathbf{1}(\hat{y}_y \leq \gamma + \max_{j \neq y} \hat{y}_j)$ and $\gamma \geq 0$ is the margin parameter. Let $\mathcal{S} = \{(x_i, y_i)\}_{i=1}^m \subseteq \mathcal{X} \times \mathcal{Y}$ denote a collection of $m$ input/label pairs sampled i.i.d. from some unknown distribution $\mathcal{D}$. Then, the empirical error of a hypothesis $g_{\mathbf{w}} : \mathcal{X} \to \mathbb{R}^K$ with parameters $\mathbf{w}$ is defined as $\hat{L}_{\mathcal{S},\gamma}(g_{\mathbf{w}}) = 1/m \sum_{i=1}^m l_\gamma(y_i, g_{\mathbf{w}}(x_i))$. Accordingly, we can define the generalization error as $L_{\mathcal{D},\gamma}(g_{\mathbf{w}}) = \mathbb{E}_{(x,y)\sim\mathcal{D}}[l_\gamma(y, g_{\mathbf{w}}(x))]$. We use $\| \cdot \|_2$ to refer the $\ell_2$ norm (vectors) and the spectral norm (matrices), and $\| \cdot \|_F$ to refer to the Frobenius norm. Also, we denote the set $\{1, ..., n\}$ by $[n]$. We provide a notation table in the Appendix (Table 5).

**Graph neural networks (GNNs).** Message-passing GNNs [15, 55] employ a sequence of message-passing steps: each node $v$ aggregates messages from its neighbors $N(v) = \{u : (v, u) \in E\}$, using the resulting vector to update its own embedding. Starting from $z_v^{(0)} = z_v$, GNNs recursively apply

$$z_v^{(\ell+1)} = \text{Upd}_\ell \left( z_v^{(\ell)}, \text{Agg}_\ell(\{\!\!\{z_u^{(\ell)} : u \in N(v)\}\!\!\}) \right) \qquad \forall v \in V, \tag{1}$$

where $\{\!\{\cdot\}\!\}$ denotes a multiset, $\mathrm{Agg}_\ell$ is an order-invariant function and $\mathrm{Upd}_\ell$ is an arbitrary update function — often a multilayer perceptron (MLP).

**Persistence homology (PH) on graphs** aims to extract detailed (multiscale) topological features from graphs. A *filtration* of a graph $G$ is a finite nested sequence of subgraphs of $G$, i.e., $\emptyset = G_0 \subset G_1 \subset ... \subset G$ — alternatively, clique complexes can also be built at each step (see [2]). While filtrations can be obtained in different ways [2, 18], a typical choice consists of leveraging a real-valued filtering function $f$ on the vertices of $G$ (or their features) to compute the vertex level set $V_\alpha = \{v : f(v) \leq \alpha\}$ at scale $\alpha \in \mathbb{R}$. Let $G_\alpha$ be the subgraph of $G$ induced by $V_\alpha$. By increasing $\alpha$ from $-\infty$ to $\infty$, we obtain a nested sequence of subgraphs called the *sub-level filtration* of $G$ induced by $f$. The idea of PH is to keep track of the appearance and disappearance of topological features (e.g., connected components, loops) in a filtration. If a topological feature first appears in $G_{\alpha_b}$ and disappears in $G_{\alpha_d}$, then we encode its persistence as a pair $(\alpha_b, \alpha_d)$; if a feature does not disappear, then its persistence is $(\alpha_b, \infty)$. The collection of all pairs forms a multiset that we call *persistence diagram*. We use $\mathrm{Dg}_i(G)$ to denote the persistence diagram for $i$-dim topological features of graph $G$. For details on PH, we refer to Edelsbrunner and Harer [10], Hensel et al. [17], and Hofer et al. [19].

**Persistence layers (PersLay).** Carrière et al. [5] introduced a general way to vectorize *persistence diagrams*. Given a persistence diagram $\mathrm{Dg}(G)$ for an arbitrary dimension, PERSLAY computes

$$\mathrm{PERSLAY}(\mathrm{Dg}(G)) = \mathrm{Agg}\left(\{\!\{\omega(p)\varphi(p) \mid p \in \mathrm{Dg}(G)\}\!\}\right) \tag{2}$$

where $\mathrm{Agg}$ is any permutation invariant operation (e.g., minimum, maximum, sum, or kth largest value), $\omega : \mathbb{R}^2 \mapsto \mathbb{R}$ is a *weight* function for the elements in $\mathrm{Dg}(G)$, and $\varphi : \mathbb{R}^2 \mapsto \mathbb{R}^h$ is the so-called *point transformation*. More specifically, given a persistence pair $p = [p_1, p_2]^\top \in \mathbb{R}^2$, PersLay introduces the *triangle point transformation* ($\Lambda$):

$$\varphi_\Lambda(p) = [\Lambda_p(t_1), ..., \Lambda_p(t_h)]^\top \quad \text{with} \quad \Lambda_p(t_i) = \max\{0, p_2 - |t_i - p_1|\}, t_i \in \mathbb{R} \tag{3}$$

the *Gaussian point transformation* ($\Gamma$):

$$\varphi_\Gamma(p) = [\Gamma_p(t_1), ..., \Gamma_p(t_h)]^\top \quad \text{with} \quad \Gamma_p(t_i) = \exp\left(-\frac{\|t_i - p\|_2^2}{2\tau^2}\right), t_i \in \mathbb{R}^2 \tag{4}$$

and the *line point transformation* ($\Psi$):

$$\varphi_\Psi(p) = [\Psi_p(t_1), ..., \Psi_p(t_h)]^\top \quad \text{with} \quad \Psi_p(t_i) = t_{i,1}p_1 + t_{i,2}p_2 + t_{i,3}, t_i \in \mathbb{R}^3 \tag{5}$$

In all of these transformations $t_1, \ldots, t_h$ are learnable parameters. Notably, the architectural design of PERSLAY is quite versatile and accommodates a wide range of traditional persistence diagram vectorizations, extending DeepSets [56] and including persistence landscapes [4], persistence silhouette [6], persistence images [1], and other Gaussian-based kernel approaches [23, 29, 31].

To obtain class predictions, the output of PersLay is typically fed to an MLP with Lipschitz activations. We refer to this joint model (PersLay followed by MLP) as *PersLay Classifier* (PC).

**GNNs with persistence.** Recently, PH has been used to boost the expressive power of GNNs. For instance, Horn et al. [20] and Immonen et al. [22] leverage node embeddings at each layer of a GNN to obtain persistence descriptors. This topological information can be added to GNN's node embeddings (as in [20]) or concatenated with GNN's graph-level representation (as in [22]) — which we refer to the

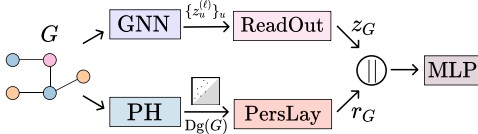

Figure 1: **GNNs with persistence** (parallel mode).

*parallel mode* of integrating PH into GNNs. Figure 1 illustrates the latter (parallel mode) with persistence diagrams vectorized using PersLay.

**PAC-Bayesian analysis** adopts a Bayesian approach to the PAC learning framework [30, 37, 38, 50]. The idea consists of placing a prior distribution $\mathcal{P}$ over our hypothesis class and then use the training data to obtain a posterior $\mathcal{Q}$, i.e., the learning process induces a posterior distribution over the hypothesis class. Importantly, we can leverage the Kullback-Leibler (KL) divergence between $\mathcal{Q}$ and $\mathcal{P}$ to bound the difference between the generalization and empirical errors [37]. To compute PAC-Bayes bounds for models like neural networks, we can i) choose a prior, ii) apply a learning algorithm; and iii) add random perturbations (from some known distribution) to the learned parameters such that we ensure tractability of the KL divergence. Following this recipe, Neyshabur et al. [40] established the following generalization bound (Lemma 1) when the change in the output of the model due to perturbations over the learned parameters is small with high probability.

**Lemma 1** (Neyshabur et al. [40]). *Let $g_{\mathbf{w}}(x) : \mathcal{X} \to \mathbb{R}^K$ be any model with parameters $\mathbf{w}$, and let $\mathcal{P}$ be any distribution on the parameters that is independent of the training data. For any $\mathbf{w}$, we construct a posterior $\mathcal{Q}(\mathbf{w} + \mathbf{u})$ by adding any random perturbation $\mathbf{u}$ to $\mathbf{w}$, s.t., $\mathbb{P}_{\mathbf{u}}(\max_{x \in \mathcal{X}} |g_{\mathbf{w}+\mathbf{u}}(x) - g_{\mathbf{w}}(x)|_{\infty} < \frac{\gamma}{4}) > \frac{1}{2}$. Then, for any $\gamma, \delta > 0$, with probability at least $1 - \delta$ over an i.i.d. size-$m$ training set $\mathcal{S}$ drawn according to $\mathcal{D}$, for any $\mathbf{w}$, we have:*

$$L_{\mathcal{D},0}(g_{\mathbf{w}}) \leq \hat{L}_{\mathcal{S},\gamma}(g_{\mathbf{w}}) + 4\sqrt{\frac{D_{KL}(\mathcal{Q}(\mathbf{w}+\mathbf{u})||\mathcal{P}) + \log\frac{6m}{\delta}}{m-1}}.$$

## 3  Generalized PAC-Bayes

This section first presents a general procedure for obtaining generalization bounds for heterogeneous models, i.e., going beyond spectrally-normalized layers and architecture-specific models (as in [33, 40]). Then, we show how to leverage such a procedure to extend existing bounds in the literature and to obtain the first generalization bound for PersLay.

Our next result (Lemma 2) applied perturbation-based PAC-Bayes bounds to arbitrary models with (possibly) non-homogeneous layers. To achieve this generality, we carefully contrasted results in [33, 40, 48] to identify the conditions (Equations 6, 7, 8, and 9) that are sufficient to subsume the considered models as well as to extend to a broader class of models. In Section 4, we will also exploit Lemma 2 in the analysis of different combinations of neural models (e.g., GNNs and PersLay).

---

**General recipe for PAC-Bayesian bounds for heterogeneous models**

**Lemma 2.** *Let $f_{\mathbf{w}} : \mathcal{X} \mapsto \mathbb{R}^K$ be a model with parameters $\mathbf{w} = vec\{W_1, ..., W_n\}$. If there exists $T \in \mathbb{R}^+$ and a sequence $(S_i)_{i \in [n]}$ with $S_i \in \mathbb{R}^+$ both of which may depend on $\mathbf{w}$, and parameter-independent $C_1, C_2 \in \mathbb{R}^+$ and sequence $(\eta_i)_{i \in [n]}$ with $\eta_i \in (0, 1]$ such that:*

- *the output is bounded by $C_1 T$:*

$$\sup_{x \in \mathcal{X}} \|f_{\mathbf{w}}(x)\|_2 \leq C_1 T, \tag{6}$$

- *the output change can be bounded under a small perturbation of the parameters, i.e., for all $\mathbf{u} = vec\{U_1, ..., U_n\}$ with $\|U_i\|_2 \leq \eta_i S_i$:*

$$\sup_{x \in \mathcal{X}} \|f_{\mathbf{w}+\mathbf{u}}(x) - f_{\mathbf{w}}(x)\|_2 \leq C_2 T \sum_{i=1}^{n} \frac{\|U_i\|_2}{S_i} \tag{7}$$

- *the following auxiliary conditions hold:*

$$\frac{1}{n}\left(\sum_{i=1}^{n} \frac{1}{S_i}\right) \geq \frac{1}{T^{1/n}}, \tag{8}$$

$$\bar{\eta} := \min_{1 \leq i \leq n} \eta_i \leq \frac{C_1}{2C_2}. \tag{9}$$

*Then, for any $\gamma, \delta > 0$ with probability at least $1 - \delta$ over the choice of training sets $\mathcal{S}$ with $m$ i.i.d. samples drawn according to some distribution $\mathcal{D}$, we have:*

$$L_{\mathcal{D},0}(f_{\mathbf{w}}) \leq \hat{L}_{\mathcal{S},\gamma}(f_{\mathbf{w}}) +$$

$$+ \mathcal{O}\left(\sqrt{\frac{\max\{1, \|\mathbf{w}\|_2^2\}T^2\left(\sum_{i=1}^{n}\frac{1}{S_i}\right)^2 h\ln(nh)\,C_1^2\bar{\eta}^{-2} + \log\max\left\{\frac{m}{\delta}, \frac{m}{\delta C_1}\right\}}{\gamma^2 m}}\right), \tag{10}$$

*where $h$ is the maximum dimension across the matrices $(W_i)_{i \in [n]}$.*

---

*Proof sketch.* We build on the main result in [40] by extending it to a broader context. The general idea involves employing Lemma 1. Following [40], we define the prior distribution $\mathcal{P}$ as an isotropic Gaussian with variance $\sigma^2$ and the posterior distribution $\mathcal{Q}$ as a shifted isotropic Gaussian with the same variance. To achieve a tighter bound, it is essential to maximize $\sigma$ (since the KL-divergence scales as $\mathcal{O}(1/\sigma^2)$); consequently, $\sigma$ should be determined based on the parameter $\beta$. However, since $\mathcal{P}$ must remain independent of the learned weights, we set $\sigma$ according to an approximation of the learned weights. Specifically, we define $\beta = T \sum_i 1/s_i$ and at first consider only $\mathbf{w}$ such that $\beta$ fall within the range $|\beta - \tilde{\beta}| \leq 1/2\beta$ for some arbitrary $\tilde{\beta}$, an approximation. We then select $\sigma$ based on this approximation, $\tilde{\beta}$. At this point we can apply Lemma 1 for all $\mathbf{w}$ such that $\beta$ falls into the defined earlier interval. To account for other values of $\beta$, we establish a finite grid across the relevant $\beta$ values and choose an appropriate $\tilde{\beta}$ for each interval on the grid. Finally, a union-bound argument across all $\tilde{\beta}$ values provides the final result. Although Equation 7 and Lemma 1 have their own constraints on the random perturbation, the above steps outline a method to set the variance $\sigma$ that satisfies these constraints and maintains independence from the learned weights. □

Table 2: **Application of Lemma 2 to MLPs and GNNs.** The detailed proof of the lemma applicability can be found in the Appendix E and the detailed description of the models in Appendix B. Here we provide brief description. We consider $n$-layer multilayer perceptron (MLP) with weights $W_1, ..., W_n$. After layer $i$ we apply $\text{Lip}_i$-Lipschitz activation function for $i \in [n-1]$. Every input is contained in $\ell_2$-ball of radius $B$. We consider $n$-layer GCN with weights $W_1, ..., W_n$. After layer $i$ we apply $\text{Lip}_i$-Lipschitz activation function. Every node feature of the graph is contained in $\ell_2$-ball of radius $B$ and the maximum degree of the node is $d - 1$. We denote $\text{Lip} = \text{Lip}_1 \cdot ... \cdot \text{Lip}_{n-1}$. We consider $n$-layer ($n > 2$) MPGNN with weights $W_1, W_2, W_3$ with activation functions $g, \phi, \rho$ with corresponding Lipschitz constants. We denote $\mathcal{C} = \text{Lip}_\phi \text{Lip}_g \text{Lip}_\rho \|W_2\|$, $\lambda = \|W_1\|_2 \|W_3\|_2$ and $\xi = ((d\mathcal{C})^{n-1}-1)/(d\mathcal{C}-1)$. Comparing to [33] we do not add $\text{Lip}_\phi$ to $\xi$ and instead of $W_l$ we have $W_3$.

| **Model** (reference) | $T$ | $S_i$ | $\eta_i$ | $C_1$ | $C_2$ |
|---|---|---|---|---|---|
| **MLP** (Neyshabur et al. [40]) | $\prod\limits_{i=1}^{n} \|W_i\|_2$ | $\|W_i\|_2$ | $\frac{1}{6n}$ | $B\,\text{Lip}$ | $eB\,\text{Lip}$ |
| **GCN** (Liao et al. [33]) | $\prod\limits_{i=1}^{n} \|W_i\|_2$ | $\|W_i\|_2$ | $\frac{1}{6n}$ | $d^{\frac{n-1}{2}} B\,\text{Lip}$ | $ed^{\frac{n-1}{2}} B\,\text{Lip}$ |
| **GCN** (Sun and Lin [48]) | $\prod\limits_{i=1}^{n} \|W_i\|_2$ | $\|W_i\|_2$ | $\frac{1}{6n}$ | $B\,\text{Lip}$ | $eB\,\text{Lip}$ |
| **MPGNN**, $d\mathcal{C} \neq 1$ (Liao et al. [33]) | $\lambda\xi$ | $\|W_i\|_2$* | $\frac{1}{6n}$ | $B\,\text{Lip}_\phi$ | $eBn\,\text{Lip}_\phi$ |
| **MPGNN**, $d\mathcal{C} = 1$ (Liao et al. [33]) | $\lambda$ | $\|W_i\|_2$* | $\frac{1}{6n}$ | $Bn\,\text{Lip}_\phi$ | $eBn^2\,\text{Lip}_\phi$ |

*$S_2 = \min\{d\mathcal{C}, \|W_2\|_2\}$

**Discussion.** We note that Lemma 2 requires choosing values for the variables $(S_i)_{i \in [n]}$ and $T$. In this regard, one might set $S_i$ as the spectral norm of the weight matrix $W_i$, i.e., $S_i = \|W_i\|_2$, and make $T$ equal to $\prod_i S_i$. In this case, the condition in Equation 8 is satisfied — the geometric mean is always smaller than or equal to the arithmetic mean. Regarding the variables $(\eta_i)_i$, a typical choice is to set $\eta_i = \mathcal{O}(1/n)$. By doing so, our bound implicitly depends on $n$ also through $\eta_i$.

The role of Equation 6 is to constraint $\mathbf{w}$ to non-trivial parameter spaces. In particular, if $T$ is too small, the magnitude of the model output might not be sufficient to distinguish different inputs up to a margin $\gamma$. In this case, the model would have large empirical loss. In turn, Equation 7 is directly associated with the condition in Lemma 1, enabling us to use it.

As discussed, Neyshabur et al. [40] assume spectrally-normalized weight matrices. To avoid this assumption, we introduce the conditions in Equation 8 and Equation 9, which allow us to pick perturbations that meet the condition in Equation 7, again justifying the application of Lemma 1.

The bound also includes a somewhat unconventional term, $\max\{1, \|\mathbf{w}\|_2^2\}$. While this technical term allows for a more concise proof, we note that it does not impose suboptimality. More specifically, in most real-world cases, the squared norm of $\mathbf{w}$ is greater than 1. See Appendix I for a discussion.

**Applying Lemma 2 to MLPs and GNNs.** As previously mentioned, using Lemma 2 involves defining the variables $T, (S_i)_i, (\eta_i)_i, C_1, C_2$ to meet all conditions in Lemma 2's statement. Typically, this definition comes naturally from the *perturbation analysis* of the model. To illustrate the power of Lemma 2, Table 2 shows how we can apply it using the perturbation analysis for MLPs and GNNs provided in [33, 40, 48]. Detailed proofs are given in Appendix E. Importantly, we note that we do not make any additional assumption beyond those in the original papers — we state all assumptions before each proof in the Appendix for clarity.

Notably, our approach generalizes results by Neyshabur et al. [40] and Liao et al. [33] to MLPs/GCNs with different activation functions — note that the original works only consider ReLU activations. For inherently non-homogeneous models like MPGNNs, our method leads to tighter bounds in several settings. We provide a comparison between our bounds and previous ones in Appendices F and G.

**Applying Lemma 2 to PersLay.** Next, Corollary 1 provides a PAC-Bayes bound for PersLay under fixed filtration functions — see Appendix H for a discussion about filtration functions. To the best of our knowledge, this is the first generalization result for vectorization schemes of persistence diagrams. Again, the proof consists of verifying if the requirements of Lemma 2 are met. For readability, we omit the proof and provide only the constant weighting function case in the main text, proof as well as the arbitrary weighting function case can be found in Appendix E. The perturbation analysis of PersLay is also given in the supplementary material (Lemma 10).

**Corollary 1** (PersLay with constant weighting function). *Let $f_{\mathbf{w}} : \mathcal{G} \mapsto \mathbb{R}^k$ with $\mathbf{w} = vec\{W^{(\varphi)}\}$ be a PersLay where $W^{(\varphi)}$ denotes the parameters of the point-transformation function, $\varphi$. Let $\mathrm{Dg}$ be a mapping from graphs to (extended) persistence diagrams with a fixed filtration function and $B$ such that $\max_{G \in \mathcal{G}} \max_{p \in \mathrm{Dg}(G)} \|p\|_2 \leq B$, then $f_{\mathbf{w}}$ satisfies the requirements of Lemma 2 with:*

$$T = T^{(\varphi)} \quad and \quad S = \|W^{(\varphi)}\|_2 \quad and \quad \eta = 1,$$

$$C_1 = 2C_2 \quad and \quad C_2 = 2A_2 \max\{Lip^{(\varphi)}, C^{(\varphi)}\},$$

*where*

$$A_1 = A_2 = \max_{G \in \mathcal{G}} card(\mathrm{Dg}(G)) \qquad if \, \mathrm{Agg} = \text{"sum"}$$
$$A_1 = 1, A_2 = 3 \qquad\qquad if \, \mathrm{Agg} = \text{"k-max" or "mean"}$$

*and*

|  | $T^{\varphi}$ | $C^{\varphi}$ | $Lip^{\varphi}$ |
|---|---|---|---|
| *if* $\varphi = \Lambda$ | $\max\{1, \|W^{(\varphi)}\|_2\}$ | $B\sqrt{h}$ | $1$ |
| *if* $\varphi = \Gamma$ | $\max\{1, \|W^{(\varphi)}\|_2\}$ | $\sqrt{h}$ | $\frac{1}{\tau\sqrt{e}}$ |
| *if* $\varphi = \Psi$ | $\|W^{(\varphi)}\|_2$ | $\sqrt{3}\max\{1, B\}$ | $\max\{1, B\}$ |

**PersLay's special cases.** Carrière et al. [5] designed PersLay with flexibility in mind to subsume commonly used persistence vectorization schemes in the literature. Consequently, we can obtain bounds for these schemes — we provide the values of $C_2$ (divided by 2) which is enough to compare the schemes since everything else is the same in the constant weighting function case:

| Persistence $k$-landscapes [4] | Persistence Images [1] | Persistence Silhouettes [6] |
|---|---|---|
| $3\max\{B\sqrt{h}, 1\}$ | $card \cdot \max\{\sqrt{h}, \frac{1}{\tau\sqrt{e}}\}$ | $card \cdot \max\{B\sqrt{h}, 1\}$ |

where $card$ is the $\max_{G \in \mathcal{G}} \mathrm{Dg}(G)$. If "images" and "silhouettes" use "sum" as an aggregating function, then our generalization analysis suggests that "$k$-landscapes" would have stronger guarantees. If these schemes use "mean" as an aggregating function, then $C_2$ for "$k$-landscapes" would be at most $C_2$ for "silhouettes", and the result of comparison of "$k$-landscapes" and "images" could be in favor of both "landscapes" and "images" depending on chosen parameters $\tau$ and $B$.

## 4 Compositional PAC-Bayes

In this section, we present two lemmas (Lemmas 3 & 4) that allow us to *compose* models satisfying Lemma 2 requirements. At the end of the section (Corollary 2), we showcase our framework by

getting generalization bounds for combinations of GNNs and PersLay. For readability, here we provide informal statements and defer the formal ones to the Appendix (Lemma 5, Lemma 6).

In particular, Lemma 3 establishes that the composition of MLPs with models that satisfy Lemma 2 also satisfy it. As a result, we can derive PAC-Bayes bounds for heterogeneous models that leverage MLPs using our framework in a straightforward way. This is particularly relevant since deep learning models often employ learnable feature extractors followed by MLPs as classification heads.

**Lemma 3** (Informal; Composition with MLP). *Let $f$ be an MLP and $g$ be a model satisfying Lemma 2 requirements, then $f \circ g$ also satisfies Lemma 2 requirements.*

In addition, we show in Lemma 5 (Appendix) that this result also extends to an arbitrary number of models beyond MLPs. In particular, the result holds whenever we can upper bound output deviations due to perturbations on parameters and inputs, i.e., $\sup_{x \in \mathcal{X}} \|f_{\mathbf{w}+\mathbf{u}}(x + \Delta x) - f_{\mathbf{w}}(x)\|_2$ is bounded.

Our next lemma suggests that models satisfying Lemma 2 requirements are closed under parallel concatenation. We note that combining two (or more) models in parallel is also a common design choice in deep learning. For instance, this encompasses persistence-augmented GNNs [22] and *ensemble* methods [13].

**Lemma 4** (Informal; Models in parallel). *Let $f_1$, $f_2$ be two models satisfying Lemma 2 requirements and $g$ be an aggregating Lipschtiz function. Then, $g(f_1(\cdot), f_2(\cdot))$ also satisfies Lemma 2 requirements.*

We also provide a generalization of this lemma in the Appendix (Lemma 6) for $n > 2$ models in parallel. It leads to tighter bounds than a naive 2-by-2 sequential application of Lemma 4.

**Corollary 2** (Informal). *By combining the results for MLPs, GCNs, MPGNNs (Table 2) and that for PersLay (Corollary 1) with Lemma 3 and Lemma 4, we can get generalization bounds on various compositions of these models. In particular, for GNNs with persistence (see Figure 1), we have*

- *$T$ for the overall model scales with the product of PersLay's and GNN's $T$ variables;*

- *$C_1$ and $C_2$ of the overall model scale linearly with $C_1, C_2$ of PersLay and GNN.*

Despite the generality of our results, Corollary 2 demonstrates the benefits of our framework in the domain of graph representational learning. To the best of our knowledge, this the first result providing generalization guarantees for graph neural networks combined with persistence vectorization schemes. Furthermore, our findings can aid practitioners in making informed architectural decisions to enhance the generalizability of their models. Specifically, in the case of combining PersLay with GNNs, a tighter bound can be achieved by selecting a PersLay dimension that is considerably smaller than the GNN dimension. Failing to do so may result in a bound dependency on the width of the form $\mathcal{O}(h\sqrt{\ln h})$ rather than $\mathcal{O}(\sqrt{h \ln h})$. Additionally, we recommend using aggregation functions such as "mean" or "$k$-max" instead of "sum" as the latter introduces a term $\max_{G \in \mathcal{G}} card(G)$ to the bound, which may be large in practical scenarios.

## 5 Related Works

**Expressivity and generalization of GNNs.** GNNs have achieved state-of-the-art performance across various applications [16, 28, 45, 51, 54], and have garnered significant attention. Maron et al. [35], Xu et al. [55] analyzed the representational power of GNNs in terms of the 1-WL test, revealing theoretical limits on their expressivity. This has motivated a surge of works aiming to go beyond 1-WL with GNNs [e.g., 32]. Regarding generalization, Scarselli et al. [46] provided upper bounds on the order of growth of VC-dimension for GNNs. Garg et al. [14] presented the first data-dependent generalization bounds for GNNs via Rademacher complexity. Recently, Morris et al. [39] employed the WL test alongside VC-dimension to gain insights about the generalization performance of GNNs. For details about the expressivity and learning of GNNs, we refer to Jegelka [24].

**Learning theory and PH.** Birdal et al. [3], Dupuis et al. [8] and Chen et al. [7] investigate connections between learning theory and topological data analysis. In particular, Birdal et al. [3], Dupuis et al. [8] explored the concept of PH dimension as a complexity measure to analyze generalization. Chen et al. [7] proposed a topological regularizer to simplify decision boundaries by penalizing non-essential topological features. In contrast, we apply learning theory to derive data-dependent generalization bounds for arbitrary heterogeneous layers, specifically targeting persistence-aware GNNs, and introduce a regularizer informed by these bounds to guide the design of robust and generalizable models.

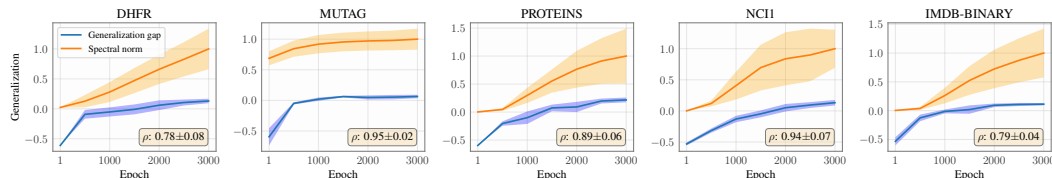

Figure 2: **PersLay classifier: spectral norm vs. generalization gap**. Overall, our bound on the spectral norm of the weights is highly correlated with the generalization gap.

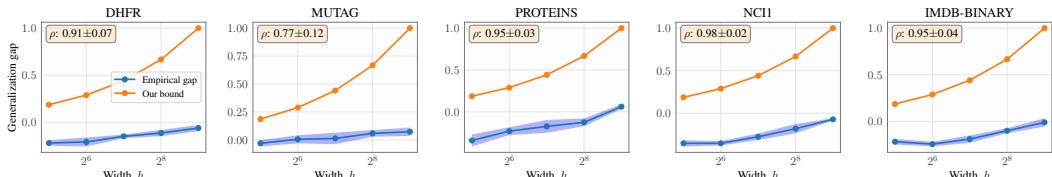

Figure 3: **PersLay classifier: width vs. generalization gap.** The dependence of the empirical gap on the model width is captured by our bound. We obtain high average correlation for all datasets.

**PAC-Bayes.** The PAC-Bayes framework [37, 38] allows us to leverage knowledge about learning algorithms and distributions over the hypothesis set to achieve tighter generalization bounds. Remarkably, Neyshabur et al. [40] presented a generalization bound for feedforward networks in terms of the product of the spectral norm of weights using a PAC-Bayes analysis. Liao et al. [33] provided PAC-Bayes bounds for GNNs, and Sun and Lin [48] enhanced their analysis considering the adversarial case as well. In a seminal work, Dziugaite and Roy [9] optimized PAC-Bayes bounds directly to obtain non-vacuous generalization bounds for deep stochastic neural networks.

## 6 Experiments

To illustrate the practical relevance of our analysis, we now consider the generalization of persistence-aware models on real-world datasets, and report results for regularized models based on our bounds. In particular, we conduct two main experiments. The first one aims to analyze how well our bounds capture generalization gaps as a function of model variables. The second assesses to which extent a structural risk minimization algorithm that uses our bound on the weights spectral norm improve generalization compared to empirical risk minimizers. We implemented all experiments using PyTorch [42], and implementation details are given in the Appendix J. Our code is available at https://github.com/Aalto-QuML/Compositional-PAC-Bayes.

**Datasets and evaluation setup.** We use six popular benchmarks for graph classification: DHFR, MUTAG, PROTEINS, NCI1, IMDB-BINARY, MOLHIV, which are available as part of TUDatasets [26] and OGB [21]. We use a 80/10/10% (train/val/test) split for all datasets when we perform model selection. Here, we consider both PersLay Classifiers and GNNs with persistence models with constant weight functions and Gaussian point transformations. For the experiments with GNNs, we kept only the larger datasets (and added results for the NCI109 dataset). Regarding filtration functions, we closely follow [5] and use Heat kernels with parameter values equal to 0.1 and 10.

**Dependence on model components**. Figure 2 and Figure 4 show the generalization gap (measured as $L_{\mathcal{D},0} - \hat{L}_{\mathcal{S},\gamma=1}$) and the bound on the weights spectral norm ($T$ from Lemma 2) over the training epochs for PersLay Classifier and GNNs with persistence, respectively. To evaluate how well our bound captures the trend observed in the empirical gap, we compute correlation coefficients between the two sequences across different seeds and report their mean and standard deviation for each dataset. Overall, the coefficients are greater than 0.7 in 7 out of 9 experiments, indicating a good correlation.

Figure 3 shows the empirical gap and our estimated bound as a function of the model's width for the PersLay classifier. Again, we compute correlation coefficients between the two curves and find they are highly correlated (with an average correlation above 0.91 on 4 out of 5 datasets). Also, we note that these curves are obtained at the final training epoch. We report additional results across different epochs and hyper-parameters in the supplementary material. Again, these results validate that our theoretical bounds can capture the trend observed in the empirical generalization gap.

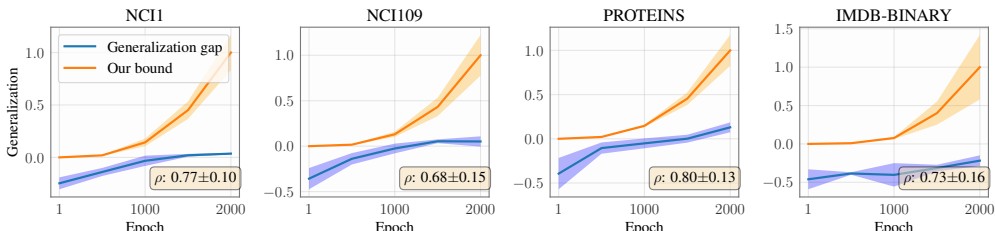

Figure 4: **GNNs with persistence: empirical gap vs. PAC-Bayes bound**. Again, there is positive and high correlation between our bound and the observed generalization gap.

**Regularizing PersLay.** We compare variants of PersLay trained via ERM (empirical risk minimization) and a regularized version with loss given by $\hat{L}_{\mathcal{S},1}+\alpha_r\sqrt{n^2\ln n\|\mathbf{w}\|_2^2\beta^2}$; $\alpha_r$ is a hyper-parameter that balances the influence of the two terms and $\beta = T\sum_i {}^1/_{S_i}$ — see proof sketch of Lemma 2 in Section 3. This is similar to a weight-decay regularization approach, with the spectral norm of weights appearing in $\beta$. Here, we consider models with $n = 1$ or $2$, selected via hold-out validation. We note that PersLay classifier does not use node features, it only exploits graph structures.

Table 3: Comparison of PersLay with and without spectral norm regularization. We report accuracy statistics (except for MOLHIV, which uses AUROC) computed over five independent runs. In 5 out of 6 cases, the models using SpecNorm achieve better test results.

| Method | DRFH | MUTAG | PROTEINS | IMDB-B | NCI1 | MOLHIV |
|---|---|---|---|---|---|---|
| PersLay (ERM) | $0.71 \pm 0.04$ | $0.88 \pm 0.02$ | $0.65 \pm 0.03$ | $0.65 \pm 0.01$ | $0.68 \pm 0.01$ | $0.7005 \pm 0.0177$ |
| PersLay (w/ SpecNorm) | $\mathbf{0.72} \pm 0.02$ | $\mathbf{0.94} \pm 0.01$ | $\mathbf{0.72} \pm 0.01$ | $\mathbf{0.70} \pm 0.01$ | $0.68 \pm 0.01$ | $\mathbf{0.7185} \pm 0.0106$ |

Table 3 reports accuracy results (mean and standard deviations) computed over five runs. Overall, the regularized approach significantly outperforms the ERM variant despite the use of small-sized networks. On 5/6 datasets, PersLay with spectral norm regularization is the best-performing model.

**Regularizing GNNs with persistence.** We now evaluate the impact of using our bound to regularize different GNNs combined with persistent homology (PersLay) in parallel mode. We consider GCN [28], GraphSage [16], and GIN [55] architectures. Table 4 reports the test classification error and the generalization gap on the NCI, NCI109, and PROTEINS datasets — mean and standard deviation obtained over five independent runs. For our regularizer, we select the optimal penalization factor $\alpha_r \in \{1e\text{-}5, 1e\text{-}6, 1e\text{-}7, 1e\text{-}8\}$ using the validation set. Overall, the results show that the regularized methods achieve smaller generalization gaps and slightly lower classification errors. In particular, our spectral regularizer leads to a significant drop in generalization gap in all experiments.

Table 4: Test classification error (0-1 loss) and generalization gap $(L_{\mathcal{D},0} - \hat{L}_{S,\gamma})$ for PH-augmented GNNs. ERM means empirical risk minimizer (no regularization). We denote the best-performing methods in bold. In almost all cases, employing the method derived from our theoretical analysis leads to the smallest test errors and generalization gaps.

| GNN | Dataset | Test error | | Generalization gap | |
|---|---|---|---|---|---|
| | | ERM | SPECNORM | ERM | SPECNORM |
| GCN | NCI | $0.22 \pm 0.01$ | $\mathbf{0.21} \pm 0.02$ | $0.19 \pm 0.01$ | $\mathbf{0.01} \pm 0.06$ |
| | NCI109 | $0.28 \pm 0.00$ | $0.28 \pm 0.02$ | $0.25 \pm 0.00$ | $\mathbf{0.12} \pm 0.03$ |
| | PROTEINS | $0.31 \pm 0.01$ | $\mathbf{0.27} \pm 0.02$ | $0.25 \pm 0.02$ | $\mathbf{-0.02} \pm 0.11$ |
| SAGE | NCI | $0.24 \pm 0.01$ | $\mathbf{0.21} \pm 0.02$ | $0.18 \pm 0.04$ | $\mathbf{-0.08} \pm 0.05$ |
| | NCI109 | $0.26 \pm 0.01$ | $\mathbf{0.24} \pm 0.01$ | $0.23 \pm 0.01$ | $\mathbf{0.05} \pm 0.06$ |
| | PROTEINS | $0.27 \pm 0.02$ | $\mathbf{0.26} \pm 0.02$ | $0.25 \pm 0.01$ | $\mathbf{-0.15} \pm 0.30$ |
| GIN | NCI | $0.25 \pm 0.01$ | $\mathbf{0.22} \pm 0.00$ | $0.23 \pm 0.01$ | $\mathbf{0.01} \pm 0.06$ |
| | NCI109 | $0.24 \pm 0.01$ | $0.24 \pm 0.03$ | $0.22 \pm 0.01$ | $\mathbf{0.00} \pm 0.08$ |
| | PROTEINS | $\mathbf{0.29} \pm 0.02$ | $0.30 \pm 0.03$ | $0.26 \pm 0.03$ | $\mathbf{0.07} \pm 0.18$ |

# 7  Conclusion, Broader Implications, and Limitations

We derive the first generalization bounds for neural networks that appeal to persistent homology for graph learning. Notably, the analyzed framework (PersLay) offers a flexible and general way to extract vector representations from persistence diagrams. Due to this generality, our analysis covers several methods available in the literature. The developed framework also allows to analyze *composite* models like, GNNs combined with PersLay. Our constructions involve a perturbation and generalization behavior analysis of **non**-homogeneous networks in rather general setting, which poses specific technical challenges.

While we provide valuable insights and methodologies, we would like to underscore the need for future investigations to delve into PH schemes that encompass parametrized filtration functions. Nonetheless, while some works showed gains using learnable filtrations [20], others have reported no benefits and advocated fixed functions instead [5, 34]. Moreover, the tightness of our bounds can further be improved since there is still considerable gap between empirical results and the theoretical one. By shedding new light on the generalization of machine learning models based on persistent homology, we hope to contribute to the community by providing key insights about the limits and power of these methods, paving the path to further theoretical developments on PH-based neural networks for graph representation learning.

## Acknowledgments

VG acknowledges support from the Research Council of Finland (grant decision 342077) for the project "Human-steered next-generation machine learning for reviving drug design", and the Jane and Aatos Erkko Foundation (grant 7001703) for "Biodesign: Use of artificial intelligence in enzyme design for synthetic biology". VG also thanks the Finnish Ministry for Education and Culture for their support via the "MEC Global Programme pilot USA" initiative. AS acknowledges the support from the Conselho Nacional de Desenvolvimento Científico e Tecnológico CNPq (404336/2023-0) and the Silicon Valley Community Foundation through the University Blockchain Research Initiative (Grant #2022-199610). KB thanks Aalto University for the support during the internship.

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

# A  Notation

Table 5 summarizes the main mathematical symbols and abbreviations used in this work.

Table 5: Summary of notation and abbreviations.

| Notation | Description |
| --- | --- |
| $G = (V, E)$ | arbitrary graph with vertices $V$ and edges $E$ |
| $\mathrm{Dg}(G)$ | persistence diagram of cardinality $card(\mathrm{Dg}(G))$ |
| $\omega(\cdot)$ | arbitrary *weight* function in the PersLay architecture, $\mathbb{R}^2 \mapsto \mathbb{R}$ |
| $W^{(\omega)}$ | parameter vector (matrix) of the $\omega$ (weight) function |
| $h$ | maximum dimensionality of the parameter vectors (matrices) |
| $\varphi(\cdot)$ | arbitrary *point transformation*, in the PersLay architecture $\mathbb{R}^2 \mapsto \mathbb{R}^h$ |
| $\varphi_\Lambda, \varphi_\Gamma, \varphi_\Psi$ | Triangle/Gaussian/Line *point transformation* |
| $t_i$ | $i$th parameter value(s) of a point transformation — following Carrière et al. [5] |
| $W^{(\varphi)}$ | vector comprising all parameters of the point transformation $\varphi$, i.e., $W^\varphi = vec(\{t_i\}_{i=1}^h)$ |
| $\mathrm{Agg}$ | arbitrary aggregation function (typically sum, mean or $k$-max) |
| PERSLAY | a mapping from *persistence diagrams* to $\mathbb{R}^h$ (Equation 2) |
| $K$ | number of classes |
| $\mathrm{Lip}_i$ | Lipschitz constant of an activation function at $i$th layer |
| $\psi_i$ | non-linear $\mathrm{Lip}_i$-Lipschitz activation function before $i$th layer |
| MLP | multi-layer perceptron with non-linear $\mathrm{Lip}_i$-Lipschitz activation functions |
| $vec(\cdot)$ | function that converts its input to a single *vector* |
| $\mathcal{D}$ | data distribution |
| $\mathcal{S}, m$ | $\mathcal{S}$ is a training set consisting of $m$ input pairs $(x, y)$ |
| $\gamma$ | margin scalar used in the margin-based loss |
| $\hat{L}_{\mathcal{S},\gamma}(g_{\mathbf{w}})$ | empirical error ($\gamma$-margin loss) of a hypothesis $g$ (with parameters $\mathbf{w}$) on $\mathcal{S}$ |
| $L_{\mathcal{D},\gamma}(g_{\mathbf{w}})$ | generalization error ($\gamma$-margin loss) of a hypothesis $g$ (with parameters $\mathbf{w}$) on $\mathcal{D}$ |
| $\ell_2$ | L-2 norm |
| $B$ | the maximum norm of the input to the with respect to $\ell_2$ norm. |
| $D_{KL}$ | KL-divergence between two distributions |
| $\|\cdot\|$ | absolute value |
| $\|\cdot\|_2, \|\cdot\|_F$ | matrix (or vector) norm induced by vector 2-norm / Frobenius norm of a matrix |
| $U$ | perturbation of the model parameter |
| $n, n_1, n_2$ | number of parameter matrices (vectors) |
| $k$ | typically the number of models we compose |
| $T$ | quantity of weights bounding max norn=m and max perturbation of a model in Lemma 2 |
| $S_i$ | *correcting* quantity appearing in max perturbation bound of a model in Lemma 2 |
| $C_1$ | constant appearing in max norm upper-bound of the model output in Lemma 2 |
| $C_2$ | constant appearing in max perturbation of a model upper-bound in Lemma 2 |
| $\eta_i, \bar{\eta}$ | maximum allowed ratio of $\|U_i\|_2$ to $\|S_i\|_2$ / min of $\eta_i$ in Lemma 2 |
| $\beta, \tilde{\beta}$ | $T \sum_{i=1}^{n} \frac{1}{S_i}$ / arbitrary approximation of $\beta$ appearing in Lemma 2 |

# B    Model descriptions

**MLP.**

$$
\begin{aligned}
H_k &= \psi_k(W_k H_{k-1}) && (k\text{-th layer}), \\
H_n &= W_n H_{n-1}, && (\text{Final layer}),
\end{aligned} \tag{11}
$$

where $H_k \in \mathbb{R}^{h_k}$ are the vectors computed after layer $k$, and $W_k \in \mathbb{R}^{h_{k-1} \times h_k}, k \in [n]$ are the parameters in the $k$-th layer, and we let $H_0 = \mathbf{x}$. $\psi_k, k \in [n-1]$ are some element-wise non-linear $\text{Lip}_k$-Lipschitz functions. Note that we change $d$ (number of layers in Neyshabur et al. [40] notation) to $n$ to be consistent with *the number of parameters* notation.

**GCN.**

$$
\begin{aligned}
H_k &= \psi_k(P_G H_{k-1} W_k) && (k\text{-th Graph Convolution Layer}), \\
H_n &= \frac{1}{|V|} \mathbf{1}_{|V|} H_{n-1} W_n && (\text{Readout layer}),
\end{aligned} \tag{12}
$$

where $H_k \in \mathbb{R}^{|V| \times h_k}, k \in [n-1]$ are the node representations in each layer, $H_n \in \mathbb{R}^{1 \times K}$ is the readout, and $W_k \in \mathbb{R}^{h_{k-1} \times h_k}, k \in [n]$ are the parameters in the $k$-th layer. And we let $H_0 = Z$, where $Z$ is the matrix constructed from node features, $z_v$ for $v \in V$. The matrix $P_G \in \mathbb{R}^{|V| \times |V|}$ is related to the graph structure and $\psi_k(\cdot), k \in [n-1]$ are some element-wise non-linear $\text{Lip}_k$-Lipschitz mappings. Practically, for GCN, we take $P_G$ as the Laplacian of the graph, defined as $\tilde{D}^{-1/2} \tilde{A} \tilde{D}^{-1/2}$, where $\tilde{A}$ is the adjacency matrix with +1 on the diagonal and $\tilde{D} = \text{diag}\left(\sum_{j=1}^{|V|} \tilde{A}_{ij}, i \in [|V|]\right)$ is the degree matrix of $\tilde{A}$. Note that we sightly changed (comparing to Liao et al. [33], Sun and Lin [48]) the notation to be consistent: instead of $l$, number of layers, we have $n$, instead of $n$, number of nodes, we have $|V|$ and instead of $\sigma_i$ we have $\psi_i$ as activation functions.

**MPGNN.**

$$
\begin{aligned}
M_k &= g(C_{\text{out}}^\top H_{k-1}) && (k\text{-th step Message Computation}) \\
\bar{M}_k &= C_{\text{in}} M_k && (k\text{-th step Message Aggregation}) \\
H_k &= \phi\left(X W_1 + \rho\left(\bar{M}_k\right) W_2\right) && (k\text{-th step Node State Update}) \\
H_n &= \frac{1}{|V|} \mathbf{1}_{|V|} H_{n-1} W_3 && (\text{Readout Layer}),
\end{aligned} \tag{13}
$$

where $k \in [n-1]$, $H_k \in \mathbb{R}^{|V| \times h_k}$ are node representations/states and $H_n \in \mathbb{R}^{1 \times K}$ is the output representation. Here we initialize $H_0 = \mathbf{0}$. WLOG, we assume $\forall k \in [n-1]$, $H_k \in \mathbb{R}^{|V| \times h}$ and $M_k \in \mathbb{R}^{|V| \times h}$ since $h$ is the maximum hidden dimension. $C_{\text{in}} \in \mathbb{R}^{|V| \times |E|}$ and $C_{\text{out}} \in \mathbb{R}^{|V| \times |E|}$ are the incidence matrices corresponding to incoming and outgoing nodes[1] respectively. Specifically, rows and columns of $C_{\text{in}}$ and $C_{\text{out}}$ correspond to nodes and edges respectively. $C_{\text{in}}[i, j] = 1$ indicates that the incoming node of the $j$-th edge is the $i$-th node. Similarly, $C_{\text{out}}[i, j] = 1$ indicates that the outgoing node of the $j$-th edge is the $i$-th node. $g, \phi, \rho$ are nonlinear mappings, e.g., ReLU and Tanh. Technically speaking, $g : \mathbb{R}^h \to \mathbb{R}^h$, $\phi : \mathbb{R}^h \to \mathbb{R}^h$, and $\rho : \mathbb{R}^h \to \mathbb{R}^h$ operate on vector-states of individual node/edge. However, since we share these functions across nodes/edges, we can naturally generalize them to matrix-states, e.g., $\tilde{\phi} : \mathbb{R}^{|V| \times h} \to \mathbb{R}^{|V| \times h}$ where $\tilde{\phi}(X)[i, :] = \phi(X[i, :])$. By doing so, the same function could be applied to matrices with varying size of the first dimension. For simplicity, we use $g, \phi, \rho$ to denote such generalization to matrices. We denote the Lipschitz constants of $g, \phi, \rho$ under the vector 2-norm as $C_g, C_\phi, C_\rho$ respectively. We also assume $g(\mathbf{0}) = \mathbf{0}$, $\phi(\mathbf{0}) = \mathbf{0}$, and $\rho(\mathbf{0}) = \mathbf{0}$ and define the *percolation complexity* as $\mathcal{C} = C_g C_\phi C_\rho \|W_2\|_2$ following Garg et al. [14]. Note that we sightly changed (comparing to Liao et al. [33], Sun and Lin [48]) the notation to be consistent: instead of $l$, number of layers, we have $n$, instead of $n$, number of nodes, we have $|V|$ and instead of $W_l$, parameter matrix, we have $W_3$.

---

[1]For undirected graphs, we convert each edge into two directed edges.

## C Proofs of main lemma and composition lemmas

**Lemma 2.** *Let $f_{\mathbf{w}} : \mathcal{X} \mapsto \mathbb{R}^K$ be a model with parameters $\mathbf{w} = vec\{W_1, ..., W_n\}$. If there exists $T \in \mathbb{R}$ and $S_i \in \mathbb{R}$ for $i \in [n]$ depending on $W_1, ..., W_n$; $C_1, C_2 > 0$; $\eta_i \in (0,1]$ for $i \in [n]$ such that:*

*maximum norm of the output is bounded by $T$ and $C_1$:*

$$\max_{x \in \mathcal{X}} \|f_{\mathbf{w}}(x)\|_2 \le C_1 T, \tag{14}$$

*maximum change of the output with perturbed weights, $\mathbf{u} = vec\{U_1, ..., U_n\}$, is bounded if the perturbation is small, $\|U_i\|_2 \le \eta_i S_i$:*

$$\max_{x \in \mathcal{X}} \|f_{\mathbf{w}+\mathbf{u}}(x) - f_{\mathbf{w}}(x)\|_2 \le C_2 T \left( \sum_{i=1}^n \frac{\|U_i\|_2}{S_i} \right), \tag{15}$$

*arithmetic mean of $S_i$ inverses is at least the $n$th root of $T$ inverse:*

$$\frac{1}{n} \left( \sum_{i=1}^n \frac{1}{S_i} \right) \ge \frac{1}{T^{1/n}}, \tag{16}$$

*minimum $\eta_i$ is at most the ratio of $C_1$ and $2C_2$:*

$$\bar{\eta} := \min_{1 \le i \le n} \eta_i \le \frac{C_1}{2C_2}, \tag{17}$$

*then for any $\gamma, \delta > 0$ with probability at least $1 - \delta$ over the choice of training set $\mathcal{S}$ of size $m$ sampled accordingly to some data distribution $\mathcal{D}$ we have the bound on the generalization gap:*

$$
\begin{aligned}
L_{\mathcal{D},0}(f_{\mathbf{w}}) \le &\hat{L}_{\mathcal{S},\gamma}(f_{\mathbf{w}})+ \\
&+ \mathcal{O}\left( \sqrt{ \frac{\max\{1, \|\mathbf{w}\|_2^2\} T^2 \left( \sum_{i=1}^n \frac{1}{S_i} \right)^2 (h \ln nh) C_1^2 \bar{\eta}^{-2} + \log \frac{m}{\delta} \max\left\{1, \frac{1}{C_1}\right\}}{\gamma^2 m} } \right),
\end{aligned}
\tag{18}
$$

*where $h$ is the upper bound for dimensions of $W_i$.*

*Proof.* First, we note that if $\mathbf{w}$ such that $C_1 T \le \frac{\gamma}{2}$, then this imply for any $i \in [K]$

$$\max_{\mathbf{x} \in \mathcal{X}} |f_{\mathbf{w}}(\mathbf{x})_i| \le \max_{\mathbf{x} \in \mathcal{X}} \|f_{\mathbf{w}}(\mathbf{x})\|_2 \le C_1 T \le \frac{\gamma}{2},$$

where we used Equation 6 and the fact that every coordinate of the vector is at most $\ell_2$-norm of the vector. In this case the empirical loss would be equal to one, since no index $i$ can satisfy $f_{\mathbf{w}}(\mathbf{x})_i \ge \gamma + \max_{i \ne j} f_{\mathbf{w}}(\mathbf{x})$ and the inequality in Equation 10 becomes trivial since generalization error can not be greater than 1.

So from now on we consider $\mathbf{w}$ such that $C_1 T \ge \frac{\gamma}{2}$. Moreover we note that:

$$C_1 T \ge \frac{\gamma}{2} \Rightarrow T \ge \frac{\gamma}{2C_1} \tag{19}$$

We denote

$$\beta := T \left( \sum_{i=1}^n \frac{1}{S_i} \right). \tag{20}$$

Following Neyshabur et al. [40] and Liao et al. [33], we consider the prior $\mathcal{P} = \mathcal{N}(\mathbf{0}, \sigma^2 \mathbf{I})$ and random perturbation $\mathbf{u} \sim \mathcal{N}(\mathbf{0}, \sigma^2 \mathbf{I})$. Note that the $\sigma$ of the prior and the perturbation are the same

and will be set according to $\beta$. More precisely, we will set the $\sigma$ based on some approximation $\tilde{\beta}$ of $\beta$ since the prior cannot depend on any learned weights directly. The approximation $\tilde{\beta}$ is chosen to be a cover set which covers the meaningful range of $\beta$. For now, let us assume that we have a fixed $\tilde{\beta}$ and consider $\beta$ which satisfies $|\beta - \tilde{\beta}| \leq \varepsilon\beta$ for some $\varepsilon > 0$. Note that this also implies:

$$|\beta - \tilde{\beta}| \leq \varepsilon\beta \Rightarrow \beta \leq \frac{1}{1-\varepsilon}\tilde{\beta} \tag{21}$$

$$|\beta - \tilde{\beta}| \leq \varepsilon\beta \Rightarrow \tilde{\beta} \leq (1+\varepsilon)\beta \tag{22}$$

This setup is very like the setup in Neyshabur et al. [40] and Liao et al. [33]

**Choosing $\sigma$.** From Tropp [49] we know that if $U \in \mathbb{R}^{h \times h}, U_{ij} \sim \mathcal{N}(0, \sigma^2)$, then

$$\mathbb{P}\left(\|U\|_2 \geq t\right) \leq 2h\exp\left(-\frac{t^2}{2h\sigma^2}\right)$$

So applying it to $U_1, ..., U_n$ (all of them are smaller than $h \times h$) and $t = \sigma\sqrt{2h\ln 4nh}$ we have:

$$\mathbb{P}\left(\|U_1\|_2 \leq \sigma C_t \ \& \ ... \ \& \ \|U_n\|_2 \leq \sigma C_t\right) \geq 1 - \sum_{i=1}^{n}\mathbb{P}\left(\|U_i\|_2 \geq \sigma C_t\right) \geq$$

$$\geq 1 - 2nh\exp\left(-\frac{\sigma^2 C_t^2}{2h\sigma^2}\right) = 1 - 2nh\exp\left(\ln\frac{1}{4nh}\right) = \frac{1}{2} \tag{23}$$

For simplicity of notation we denote $C_t = \sqrt{2h\ln 4nh}$.

Using defined perturbation $\mathbf{u} = vec\{U_1, ..., U_n\}$ and Equation 23 combined with Equation 7 we get with probability at least $\frac{1}{2}$:

$$\max_{x \in \mathcal{X}}\|f_{\mathbf{w}+\mathbf{u}}(x) - f_{\mathbf{w}}(x)\|_\infty \leq \max_{x \in \mathcal{X}}\|f_{\mathbf{w}+\mathbf{u}}(x) - f_{\mathbf{w}}(x)\|_2 \leq$$

$$\leq C_2 T\left(\sum_{i=1}^{n}\frac{\|U_i\|_2}{S_i}\right) \leq$$

$$\leq C_2\beta\sigma C_t \leq \tag{24}$$

$$\leq C_2 C_t\frac{\sigma\tilde{\beta}}{1-\varepsilon}$$

If

$$\sigma \leq \frac{\gamma(1-\varepsilon)}{4C_2 C_t\tilde{\beta}},$$

then the perturbation is bounded by $\frac{\gamma}{4}$. However, we need to satisfy the condition of Equation 7, i.e. $\forall i \in [n]: \|U_i\| \leq \eta_i S_i$. Let us find $C_\sigma \geq 1$ such that:

$$\sigma := \frac{\gamma(1-\varepsilon)}{4C_2 C_\sigma C_t\tilde{\beta}} \leq \frac{\gamma(1-\varepsilon)}{4C_2 C_t\tilde{\beta}} \tag{25}$$

and that

$$\forall i \in [n]: \ \sigma C_t \leq \eta_i S_i$$

$$\sigma C_t = \frac{\gamma(1-\varepsilon)}{4C_2 C_\sigma \tilde{\beta}} \overset{(i)}{\leq} \frac{\gamma}{4C_2 C_\sigma \beta} \overset{(ii)}{=}$$

$$\overset{(ii)}{=} \frac{\gamma \eta_i S_i}{4C_2 C_\sigma \eta_i S_i \beta} \overset{(iii)}{=} \frac{\gamma \, \eta_i S_i}{4C_2 C_\sigma \eta_i T \left(1 + \sum_{k \neq i} \frac{S_i}{S_k}\right)} \overset{(iv)}{\leq}$$

$$\overset{(iv)}{\leq} \frac{\gamma \, \eta_i S_i}{4C_2 C_\sigma \eta_i T} \overset{(v)}{\leq} \frac{\gamma \, \eta_i S_i}{4C_2 C_\sigma \eta_i \frac{\gamma}{2C_1}} =$$

$$= \eta_i S_i \frac{C_1}{2C_2 C_\sigma \, \eta_i}. \tag{26}$$

(i) comes from Equation 21. (ii) we just multiplied and divided on $\eta_i S_i$. (iii) we plugged in the expression for $\beta$, Equation 20. (iv) $1 + \sum_{k \neq i} \frac{S_i}{S_k} \geq 1$. (v) comes from Equation 19

We choose

$$C_\sigma := \frac{C_1}{2C_2 \min_{1 \leq i \leq n} \eta_i} = \frac{C_1}{2C_2 \, \bar{\eta}}. \tag{27}$$

In this case $C_\sigma \geq 1$ because of Equation 9 and continuing Equation 26:

$$\sigma C_t \leq \eta_i S_i \frac{C_1}{2C_2 C_\sigma \eta_i} = \eta_i S_i \frac{\bar{\eta}}{\eta_i} \overset{(i)}{\leq} \eta_i S_i,$$

where (i) comes from the fact that $\bar{\eta}$ is the minimum among $\eta_i$s, gives us that $\forall i \in [n] : \|U_i\|_2 \leq \eta_i S_i$.

Plugging this into expression for $\sigma$, Equation 25, we get:

$$\sigma = \frac{\gamma(1-\varepsilon)}{4C_2 C_\sigma \tilde{\beta}} = \frac{\gamma(1-\varepsilon)\bar{\eta}}{2C_1 C_t \tilde{\beta}} \tag{28}$$

At this point, we selected a prior, $\mathcal{P} = \mathcal{N}(\mathbf{0}, \sigma^2 \mathbf{I})$, and perturbation distribution, $\mathcal{Q} = \mathcal{N}(\mathbf{w}, \sigma^2 \mathbf{I})$, such that the conditions of Lemma 1 can be satisfied.

**Getting the bound for weights** $|\beta - \tilde{\beta}| \leq \varepsilon\beta$. Now, we can compute the KL-divergence and get the bound using Lemma 1 for case when $\beta$ is *around* $\tilde{\beta}$, i.e. $\mathbf{w}$ such that $|\beta - \tilde{\beta}| \leq \varepsilon\beta$.

$$D_{KL}(\mathcal{Q}(\mathbf{w} + \mathbf{u}) \| \mathcal{P}) = \frac{\|\mathbf{w}\|_2^2}{2\sigma^2} \overset{(i)}{=} \frac{\|\mathbf{w}\|_2^2}{2} \frac{4C_1^2 C_t^2 \tilde{\beta}^2}{\gamma^2 (1-\varepsilon)^2 \bar{\eta}^2} \overset{(ii)}{\leq}$$

$$\overset{(ii)}{\leq} 2\|\mathbf{w}\|_2^2 \frac{C_1^2 C_t^2 \left(\frac{1+\varepsilon}{1-\varepsilon}\right)^2 \beta^2}{\gamma^2 \bar{\eta}^2} \tag{29}$$

where (i) comes from $\sigma$ definition, Equation 28, (ii) comes from the upper-bound on $\tilde{\beta}$, Equation 22.

We are ready to put this into Lemma 1:

$$L_{\mathcal{D},0}(f_{\mathbf{w}}) \leq \hat{L}_{\mathcal{S},\gamma}(f_{\mathbf{w}}) + 4\sqrt{\frac{D_{KL}(\mathcal{Q}(\mathbf{w} + \mathbf{u}) \| \mathcal{P}) + \log \frac{6m}{\delta}}{m-1}} =$$

$$= \hat{L}_{\mathcal{S},\gamma}(f_{\mathbf{w}}) + \mathcal{O}\left(\sqrt{\frac{\max\{1, \|\mathbf{w}\|_2^2\}\beta^2 C_1^2 C_t^2 \left(\frac{1+\varepsilon}{1-\varepsilon}\right)^2 \bar{\eta}^{-2} + \ln \frac{m}{\delta}}{\gamma^2 m}}\right), \tag{30}$$

where we upper-bounded $\|\mathbf{w}\|_2^2$ as $\max\{1, \|\mathbf{w}\|_2^2\}$ to simplify further derivations.

**Union bound.**   Finally, we need to consider multiple choices of $\tilde{\beta}$ so that for any $\beta$, we can bound the generalization error like Equation 30. In order to do this we (i) find interval of reasonable $\beta$s, (ii) define a covering and upper-bound the number of balls in the covering, (iii) combine bounds for every $\tilde{\beta}$ together and get the final result.

If $\beta$ is too large, the KL-divergence would be too large and generalization gap would be greater than 1 which thereby trivialize the bound since the generalization error cannot be greater than 1.

$$
\sqrt{\frac{\max\{1, \|\mathbf{w}\|_2^2\}\beta^2 C_1^2 C_t^2 \left(\frac{1+\varepsilon}{1-\varepsilon}\right)^2 \bar{\eta}^{-2} + \ln \frac{m}{\delta}}{\gamma^2 m}} \overset{(i)}{\geq}
$$

$$
\overset{(i)}{\geq} \sqrt{\frac{\beta^2 C_1^2}{\gamma^2 m \bar{\eta}^2}} \overset{(ii)}{\geq} \frac{C_1 \beta}{\gamma \sqrt{m}}, \tag{31}
$$

where (i) holds since we remove multipliers that are greater than 1 and $\ln \frac{m}{\delta}$ which is $\geq 0$ and appears in sum, (ii) comes from the fact that $\bar{\eta} \leq 1$.

So, if $\beta \geq \frac{\gamma \sqrt{m}}{C_1}$, then the bound becomes trivial.

On the other hand, if the $\beta$ is too small, then the empirical loss would be too large since the model would not be able to classify with *margin*, $\gamma$, as it was shown in Equation 19. Using this equation and Arithmetic versus Geometric mean we get the following inequality:

$$
\beta = T \sum_{i=1}^{n} \frac{1}{S_i} \overset{(i)}{\geq} nT^{\frac{n-1}{n}} \overset{(ii)}{\geq} n\left(\frac{\gamma}{2C_1}\right)^{\frac{n-1}{n}}, \tag{32}
$$

where (i) comes from Equation 8 and (ii) comes from Equation 19.

Combining lower (Equation 32) and upper (Equation 31) bound on $\beta$ we get:

$$
n\left(\frac{\gamma}{2C_1}\right)^{\frac{n-1}{n}} \leq \beta \leq \frac{\gamma \sqrt{m}}{C_1} \tag{33}
$$

To satisfy the condition $|\beta - \tilde{\beta}| \leq \varepsilon \beta$ we can take $\varepsilon n \left(\frac{\gamma}{2C_1}\right)^{\frac{n-1}{n}}$ as the radius of the covering, $C$. In this case let us derive the upper bound for the size of the covering, $|C|$:

$$
|C| \leq \frac{\sqrt{m}}{\varepsilon n}\left(\frac{\gamma}{2C_1}\right)^{\frac{1}{n}} \leq 2\sqrt{m}\frac{1}{n}\left(\frac{1}{C_1}\right)^{1/n} \tag{34}
$$

Let us consider cases:

$$
\begin{cases}
\frac{1}{n}\left(\frac{1}{C_1}\right)^{1/n} \leq \frac{1}{C_1}, & C_1 < 1 \\
\frac{1}{n}\left(\frac{1}{C_1}\right)^{1/n} \leq 1, & C_1 \geq 1
\end{cases}
$$

First inequality holds because function $\frac{a^{1/x}}{x}$ is monotonously decreasing on the interval $(0, +\infty)$. So, we can combine:

$$
|C| \leq 2\sqrt{m} \max\left\{1, \frac{1}{C_1}\right\} \tag{35}
$$

It is left to apply the union bound argument for the events of Equation 30 happening with $\tilde{\beta}$ taking value from $i$th ball. Let us denote such event $E_i$:

$$
\mathbb{P}\left(E_1 \ \& \ ... \ \& \ E_{|C|}\right) = 1 - \mathbb{P}(\exists i : \ \text{not } E_i) \geq 1 - \sum_{i=1}^{|C|} \mathbb{P}(\text{not } E_i) \geq 1 - |C|\delta' \tag{36}
$$

Hence, if we set $\delta' = \frac{\delta}{|C|}$, then with probability at least $1 - \delta$ for all $\mathbf{w}$ we have:

$$L_{\mathcal{D},0}(f_{\mathbf{w}}) \leq \hat{L}_{\mathcal{S},\gamma}(f_{\mathbf{w}})+$$

$$+ \mathcal{O}\left( \sqrt{\frac{\max\{1, \|\mathbf{w}\|_2^2\} T^2 \left( \sum\limits_{i=1}^{n} \frac{1}{S_i} \right)^2 C_t^2 C_1^2 \bar{\eta}^{-2} + \log \frac{m}{\delta} \max\left\{1, \frac{1}{C_1}\right\}}{\gamma^2 m}} \right)$$

Note that we substituted $\varepsilon$ with $\frac{1}{2}$ and now $\max\{1, \frac{1}{C_1}\}$ appearing under logarithm comparing to Equation 30. $\square$

**Lemma 5** (Generalization of Lemma 3). *Let $f_{\mathbf{w}_1}^{(1)} : \mathcal{X}_1 \mapsto \mathbb{R}^{h_1}, ..., f_{\mathbf{w}_k}^{(k)} : \mathcal{X}_k \mapsto \mathbb{R}^{h_k}$ for $k \in \mathbb{N}$ with $\mathbf{w}_i = vec\{W_1^{(i)}, ..., W_{n_i}^{(i)}\}$ and $h_1 = K$ be some models that we can **compose**, i.e. $f = f^{(1)}(f^{(2)}(...(f^{(k)})...))$. If there exists for $j \in [k] : T^{(j)}$ depending on $W_1^{(j)}, ..., W_{n_j}^{(j)}$; for $j \in [j] : S_1^{(j)}, ..., S_{n_j}^{(1)} \in \mathbb{R}$ depending on $W_1^{(j)}, ..., W_{n_j}^{(j)}$; $\eta_1^{(1)}, ..., \eta_{n_1}^{(1)}, ..., \eta_1^{(k)}, ..., \eta_{n_k}^{(k)} \in (0, 1]$ and $C_1^{(1)}, ..., C_1^{(k)} \in \mathbb{R}$, $C_2^{(1)}, ..., C_2^{(k)} \in \mathbb{R}$ such that:*

*maximum output for every model is bounded as follows:*

$$\forall j \in [k], \forall \mathbf{x} \in \mathcal{X}_j : \ \|f_{\mathbf{w}_j}^{(j)}(\mathbf{x})\|_2 \leq C_1^{(j)} T^{(j)} \|x\|_2, \tag{37}$$

*and for $j = k$ we can have weaker condition – the same as in Lemma 2,*

*maximum perturbation of the output during small perturbation of weights, $\|U_i^{(j)}\|_2 \leq \eta_i^{(j)} S_i^{(j)}$, for every model is bounded as follows:*

$$\forall j \in [k], \forall \mathbf{x}, \Delta\mathbf{x} \in \mathcal{X}_j : \ \|f_{\mathbf{w}_j + \mathbf{u}_j}(\mathbf{x} + \Delta\mathbf{x}) - f_{\mathbf{w}_j}(\mathbf{x})\|_2 \leq$$

$$\leq C_2^{(j)} T^{(j)} \left( \|\Delta\mathbf{x}\|_2 + \|\mathbf{x}\|_2 \sum_{i=1}^{n_j} \frac{\|U_i^{(j)}\|_2}{\|S_i^{(j)}\|_2} \right), \tag{38}$$

*and for $j = k$ we can have weaker condition – the same as in Lemma 2,*

*arithmetic mean of inverses of $S_i^{(j)}$ is greater than $n_j$-root of $T^{(j)}$:*

$$\forall j \in [k] : \ \frac{1}{n_j} \sum_{i=1}^{n_j} \frac{1}{S_i^{(j)}} \geq \left( \frac{1}{T^{(j)}} \right)^{1/n_j}, \tag{39}$$

*and $\bar{\eta}^{(j)} := \min\limits_{i \in [n_j]} \eta_i^{(j)}$ is upper-bounded by the ration $C_1^{(j)}$ and $C_2^{(j)}$:*

$$\forall j \in [k] : \ \frac{C_1^{(j)}}{2C_2^{(j)}} \geq \bar{\eta}^{(j)}, \tag{40}$$

*we denote $n = \sum\limits_{j=1}^{k} n_j$, then $f$ meets requirements of Lemma 2 with*

$$T = \prod_{i=1}^{k} T^{(i)} \quad and \quad S_{1:n} = S_1^{(1)}, ..., S_{n_1}^{(1)}, ..., S_1^{(k)}, ..., S_{n_k}^{(k)}$$

$$\eta_{1:n} = \eta_1^{(1)}, ..., \eta_{n_1}^{(1)}, ..., \eta_1^{(k)}, ..., \eta_{n_k}^{(k)}$$

$$C_1 = \max_{\mathbf{x} \in \mathcal{X}_k} \|\mathbf{x}\|_2 \max \left\{ \prod_{j=1}^{k} C_1^{(j)}, C \cdot \frac{C_1^{(\text{ind})}}{C_2^{(\text{ind})}} \right\}$$

$$C_2 = \max_{\mathbf{x} \in \mathcal{X}_k} \|\mathbf{x}\|_2 C,$$

*where* $\text{ind} = \arg \min_{j \in [k]} \bar{\eta}^{(j)}$ *and* $C = \max_{j \in [k]} \prod_{i=1}^{j} C_2^{(i)} \prod_{i=j+1}^{k} C_1^{(i)}$

*Proof.* We denote as $f^{k:j}$ composition of models from $k$ to $j$

First we test Equation 6 by plugging in Equation 37 for every model:

$$\max_{\mathbf{x} \in \mathcal{X}_k} \|f^{k:1}(\mathbf{x})\|_2 \leq \max_{\mathbf{x} \in \mathcal{X}_k} C_1^{(1)} T^{(1)} \|f^{k:2}(\mathbf{x})\|_2 \leq$$

$$\leq \max_{\mathbf{x} \in \mathcal{X}_k} \|\mathbf{x}\|_2 \prod_{j=1}^{k} C_1^{(i)} T^{(i)} \leq C_1 T \tag{41}$$

Next, we test Equation 7 by plugging in Equation 38 and Equation 37 for every model:

$$\max_{\mathbf{x} \in \mathcal{X}_k} \|f_{\mathbf{w}+\mathbf{u}}^{k:1}(\mathbf{x}) - f_{\mathbf{w}}^{k:1}(\mathbf{x})\|_2 \leq$$

$$\leq \max_{\mathbf{x} \in \mathcal{X}_k} C_2^{(1)} T^{(1)} \left( \|f_{\mathbf{w}+\mathbf{u}}^{k:2}(\mathbf{x}) - f_{\mathbf{w}}^{k:2}\|_2 + \|f_{\mathbf{w}}^{k:2}(\mathbf{x})\|_2 \left( \sum_{i=1}^{n_1} \frac{\|U_i^{(1)}\|_2}{S_i^{(1)}} \right) \right) \leq$$

$$\leq \max_{\mathbf{x} \in \mathcal{X}_k} \|\mathbf{x}\|_2 \prod_{j=1}^{k} T^{(j)} \left( \sum_{j=1}^{k} \prod_{i=1}^{j} C_2^{(i)} \prod_{i=j+1}^{(k)} C_1^{(i)} \sum_{i=1}^{n_j} \frac{\|U_i^{(j)}\|_2}{S_2^{(j)}} \right) \leq \tag{42}$$

$$\leq C_2 T \sum_{j=1}^{k} \sum_{i=1}^{n_j} \frac{\|U_i^{(j)}\|_2}{S_i^{(j)}}$$

To test Equation 8 we use arithmetic versus geometric mean inequality:

$$T \sum_{j=1}^{k} \sum_{i=1}^{n_j} \frac{1}{S_i^{(j)}} \geq \sum_{j=1}^{k} n_j \left( T^{(j)} \right)^{\frac{n_j-1}{n_j}} \prod_{i \neq j} T^{(i)} \geq$$

$$\geq n \left( \prod_{j=1}^{k} \left( \left( T^{(j)} \right)^{\frac{n_j-1}{n_j}} \prod_{i \neq j} T^{(i)} \right)^{n_j} \right)^{1/n} \geq \tag{43}$$

$$\geq n T^{\frac{n-1}{n}}$$

It is left to verify Equation 9. As one can notice $\frac{C_1}{2C_2} \geq \frac{C_1^{(\text{ind})}}{C_2^{(\text{ind})}} \geq \bar{\eta}$. $\qquad \square$

**Lemma 6** (Generalization of Lemma 4). *Let* $f_{\mathbf{w}_1}^{(1)} : \mathcal{X} \mapsto \mathbb{R}^{h_1}, ..., f_{\mathbf{w}_k}^{(k)} : \mathcal{X} \mapsto \mathbb{R}^{h_k}$ *with* $\mathbf{w}_1 = vec\{W_1^{(1)}, ..., W_{n_1}^{(1)}\}, ..., \mathbf{w}_k = vec\{W_1^{(k)}, ..., W_{n_k}^{(k)}\}$ *be some models satisfying Lemma 2 conditions. If* $\text{Agg} : \mathbb{R}^{h_1} \times ... \times \mathbb{R}^{h_k} \mapsto \mathbb{R}^K$ *such that:*

$$\forall \mathbf{x}_1, \mathbf{y}_1 \in \mathbb{R}^{h_1}, ..., \mathbf{x}_k, \mathbf{y}_k \in \mathbb{R}^{h_k} : \|\text{Agg}(\mathbf{x}_1, ..., \mathbf{x}_k) - \text{Agg}(\mathbf{y}_1, ..., \mathbf{y}_k)\|_2 \leq A \sum_{j=1}^{k} \|\mathbf{x}_j - \mathbf{y}_j\|_2$$

*for some $A > 0$ and $\mathrm{Agg}(\mathbf{0}, ..., \mathbf{0}) = \mathbf{0}$. Also we denote $n = \sum_{j=1}^{k} n_j$, then $\mathrm{Agg}(f_{\mathbf{w}_1}^{(1)}(\cdot), ..., f_{\mathbf{w}_k}^{(k)}(\cdot))$ satisfies Lemma 2 conditions with either:*

$$T = \max\left\{T^{(1)}, ..., T^{(k)}, \prod_{j=1}^{k} T^{(j)}\right\} \quad and \quad S_{1:n} = S_1^{(1)}, ..., S_{n_1}^{(1)}, ..., S_1^{(k)}, ..., S_{n_k}^{(k)}$$

$$\eta_{1:n} = \eta_1^{(1)}, ..., \eta_{n_1}^{(1)}, ..., \eta_1^{(k)}, ..., \eta_{n_k}^{(k)}$$

$$C_1 = A \max\left\{\sum_{j=1}^{k} C_1^{(j)}, \max_{j \in [k]} C_2^{(j)} \cdot \frac{C_1^{(\mathrm{ind})}}{C_2^{(\mathrm{ind})}}\right\}$$

$$C_2 = A \max_{j \in [k]} C_2^{(j)},$$

*or with*

$$T = \max\left\{\sum_{j=1}^{k} T^{(j)}, \prod_{j=1}^{k} T^{(j)}\right\} \quad and \quad S_{1:n} = S_1^{(1)}, ..., S_{n_1}^{(1)}, ..., S_1^{(k)}, ..., S_{n_k}^{(k)}$$

$$\eta_{1:n} = \eta_1^{(1)}, ..., \eta_{n_1}^{(1)}, ..., \eta_1^{(k)}, ..., \eta_{n_k}^{(k)}$$

$$C_1 = A \max\left\{\max_{j \in [k]} C_1^{(j)}, \max_{j \in [k]} C_2^{(j)} \cdot \frac{C_1^{(\mathrm{ind})}}{C_2^{(\mathrm{ind})}}\right\}$$

$$C_2 = A \max_{j \in [k]} C_2^{(j)},$$

*where* $\mathrm{ind} = \arg\min_{j \in [k]} \bar{\eta}^{(j)}$.

*Proof.* First, we test Equation 6 for composite model by applying assumption about Agg and its Equation 6 for every $f^{(j)}$:

$$\max_{\mathbf{x} \in \mathcal{X}} \|\mathrm{Agg}(f_{\mathbf{w}_1}^{(1)}(\mathbf{x}), ..., f_{\mathbf{w}_k}^{(k)}(\mathbf{x}))\| \le A \sum_{j=1}^{k} \|f_{\mathbf{w}_j}^{(j)}(\mathbf{x})\|_2 \le \begin{cases} A \sum_{j=1}^{k} T^{(j)} \max_{j \in [k]} C_1^{(j)} \\ A \max_{j \in [k]} T^{(j)} \sum_{j=1}^{k} C_1^{(j)} \end{cases} \tag{44}$$

To test Equation 7 for composite model we again apply assumption about aggregation function and its Equation 7 version for every $f^{(j)}$:

$$\max_{\mathbf{x} \in \mathcal{X}} \|\mathrm{Agg}(f_{\mathbf{w}_1 + \mathbf{u}_1}^{(1)}(\mathbf{x}), ..., f_{\mathbf{w}_k + \mathbf{u}_k}^{(k)}(\mathbf{x})) - \mathrm{Agg}(f_{\mathbf{w}_1}^{(1)}(\mathbf{x}), ..., f_{\mathbf{w}_k}^{(k)}(\mathbf{w}))\|_2 \le$$

$$\le A \sum_{j=1}^{k} \|f_{\mathbf{w}_j + \mathbf{u}_j}^{(j)}(\mathbf{x}) - f_{\mathbf{w}_j}^{(j)}(\mathbf{x})\|_2 \le A \sum_{j=1}^{k} C_2^{(j)} T^{(j)} \sum_{i=1}^{n_j} \frac{\|U_i^{(j)}\|_2}{S_i^{(j)}} \le$$

$$\le AT \max_{j \in [k]} C_2^{(j)} \sum_{j=1}^{k} \sum_{i=1}^{n_i} \frac{\|U_i^{(j)}\|}{S_i^{(j)}} \tag{45}$$

To test Equation 8 for composite model we employ arithmetic vs geometric mean together with Equation 8 for every model:

$$T \sum_{j=1}^{k} \sum_{i=1}^{n_j} \frac{1}{S_i^{(j)}} = \sum_{j=1}^{k} \frac{T}{T^{(j)}} \sum_{i=1}^{n_j} \frac{T^{(j)}}{S_i^{(j)}} \ge \sum_{j=1}^{k} n_j \frac{T}{(T^{(j)})^{1/n_j}} \ge n \frac{T}{\left(\prod_{j=1}^{k} T^{(j)}\right)^{1/n}} \ge nT^{\frac{n-1}{n}}, \tag{46}$$

where the last inequality comes from the fact that $\prod_{j=1}^{k} T(j) \leq T$

It is left to test Equation 9. As one can notice in either case $\frac{C_1}{2C_2} \geq \frac{C_1^{(\text{ind})}}{2C_2^{(\text{ind})}} \geq \bar{\eta}$

$\square$

## D Perturbation analysis

**Lemma 7.** *Let $\mathbf{w} = vec\{W_1, ..., W_n\}$ be a vector of weight matrices of an $n$-layer MLP, $f_{\mathbf{w}} : \mathcal{X} \mapsto \mathbb{R}^K$. Let $\psi_i$ be a $Lip_i$-Lipschitz activation function after layer $i$, for $i \in [n-1]$. Then for any input and input perturbation, $\mathbf{x}, \Delta\mathbf{x} \in \mathcal{X}$, weight perturbation, $\mathbf{u} = vec\{U_1, ..., U_n\}$, and constants, $\eta_1, ..., \eta_n$, such that for $i \in [n] :\ \|U_i\|_2 \leq \eta_i \|W_i\|_2$, we have two inequalities:*

$$\|f_{\mathbf{w}}(\mathbf{x})\|_2 \leq \left( \prod_{i=1}^{n} Lip_i \|W_i\|_2 \right) |\mathbf{x}|_2 \tag{47}$$

$$\|f_{\mathbf{w}+\mathbf{u}}(\mathbf{x}+\Delta\mathbf{x}) - f_{\mathbf{w}}(\mathbf{x})\|_2 \leq \left( \prod_{i=1}^{n} 1+\eta_i \right) \prod_{i=1}^{n} Lip_i \|W_i\|_2 \left( |\Delta\mathbf{x}|_2 + |\mathbf{x}|_2 \sum_{i=1}^{n} \frac{\|U_i\|_2}{\|W_i\|_2} \right), \tag{48}$$

*where we denote $Lip_n = 1$.*

*Proof.* The proof follows the Neyshabur et al. [40] with some modifications concerning the $\Delta\mathbf{x}$ and $Lip_i$ and $\eta_i$.

We denote truncated $f$ after $i+1$th layer as $f^{(i+1)}$. First, we provide the bound on the norm of the output after $i+1$th layer:

$$\|f_{\mathbf{w}}^{(i+1)}(\mathbf{x})\|_2 \overset{(i)}{=} \|W_{i+1}(\psi_i(f_{\mathbf{w}}^{(i)}(\mathbf{x}))\|_2 \overset{(ii)}{\leq} \|W_{i+1}\|_2 \|\psi_i(f_{\mathbf{w}}^{(i)}(\mathbf{x}))\|_2 \overset{(iii)}{\leq}$$
$$\overset{(iii)}{\leq} \|W_{i+1}\|_2 Lip_i \|f_{\mathbf{w}}^{(i)}(\mathbf{x})\|_2, \tag{49}$$

where (i) is the definition of MLP, (ii) comes from the definition of operator norm, (iii) comes from Lipschitzness. Unrolling the recursion of Equation 49 will get us (we denote $f_{\mathbf{w}}^{(0)}(\mathbf{x})$ to be $\mathbf{x}$):

$$\|f_{\mathbf{w}}^{(i+1)}(\mathbf{x})\|_2 \leq \left( \prod_{k=1}^{i+1} \|W_k\|_2 \right) \left( \prod_{k=1}^{i} Lip_k \right) |\mathbf{x}|_2 \tag{50}$$

Now let us provide the bound for the perturbation after $i+1$th layer. We denote this perturbation as $\Delta_{i+1}$:

$$\Delta_{i+1} \overset{(i)}{=} \left\| f_{\mathbf{w}+\mathbf{u}}^{(i+1)}(\mathbf{x}+\Delta\mathbf{x}) - f_{\mathbf{w}}^{(i+1)}(\mathbf{x}) \right\|_2 \overset{(ii)}{=}$$

$$\overset{(ii)}{=} \left\| (W_{i+1}+U_{i+1})\psi_i(f_{\mathbf{w}+\mathbf{u}}^{(i)}(\mathbf{x}+\Delta\mathbf{x})) - W_{i+1}\psi_i((f_{\mathbf{w}}^{(i)}(\mathbf{x}))) \right\|_2 \overset{(iii)}{=}$$

$$\overset{(iii)}{=} \left\| (W_{i+1}+U_{i+1})(\psi_i(f_{\mathbf{w}+\mathbf{u}}^{(i)}(\mathbf{x}+\Delta\mathbf{x})) - \psi_i(f_{\mathbf{w}}^{(i)}(\mathbf{x}))) + U_{i+1}\psi_i(f_{\mathbf{w}}^{(i)}(\mathbf{x})) \right\|_2 \overset{(iv)}{\leq}$$

$$\overset{(iv)}{\leq} \|W_{i+1}+U_{i+1}\|_2 \|\psi_i(f_{\mathbf{w}+\mathbf{u}}^{(i)}(\mathbf{x}+\Delta\mathbf{x})) - \psi_i(f_{\mathbf{w}}^{(i)}(\mathbf{x}))\|_2 + \|U_{i+1}\|_2 \|\psi_i(f_{\mathbf{w}}^{(i)}(\mathbf{x}))\|_2 \overset{(v)}{\leq}$$

$$\overset{(v)}{\leq} (1+\eta_{i+1})\|W_{i+1}\|_2 Lip_i \Delta_i + \|U_{i+1}\|_2 \|\mathbf{x}\|_2 \prod_{k=1}^{i} Lip_k \prod_{k=1}^{i} \|W_k\|_2 \overset{(vi)}{=}$$

$$\overset{(vi)}{=} (1+\eta_{i+1})\|W_{i+1}\|_2 Lip_i \Delta_i + |\mathbf{x}|_2 \left( \prod_{k=1}^{i+1} \|W_k\|_2 \right) \left( \prod_{k=1}^{i} Lip_k \right) \frac{\|U_{i+1}\|_2}{\|W_{i+1}\|_2}$$

$$\tag{51}$$

where (i) comes from the definition of $\Delta_{i+1}$, (ii) comes from the defintion of an MLP, (iii) add and substract $U_{i+1}\psi_i(f_{\mathbf{w}}^{(i)}(\mathbf{x}))$, (iv) comes from triangle inequality combined with operator norm definition, (v) comes from the definition of $\eta_{i+1}$, Lipschitzness and Equation 50, (vi) comes from multiplying and dividing by $\|W_{i+1}\|_2$ the second term. Now we unroll the recursion of Equation 51

$$
\begin{aligned}
\Delta_{i+1} &\overset{(i)}{\leq} (1+\eta_{i+1})\|W_{i+1}\|_2 \mathrm{Lip}_i \Delta_i + \|\mathbf{x}\|_2 \left(\prod_{k=1}^{i+1}\|W_k\|_2\right)\left(\prod_{k=1}^{i}\mathrm{Lip}_k\right)\frac{\|U_{i+1}\|_2}{\|W_{i+1}\|_2} \overset{(ii)}{\leq} \\
&\overset{(ii)}{\leq} \left(\prod_{k=1}^{i+1}1+\eta_k\right)\left(\prod_{k=1}^{i+1}\|W_k\|_2\right)\left(\prod_{k=1}^{i}\mathrm{Lip}_k\right)\|\Delta\mathbf{x}\|_2 + \|\mathbf{x}\|_2\left(\prod_{k=1}^{i+1}\|W_k\|_2\right)\left(\prod_{k=1}^{i}\mathrm{Lip}_k\right)\left(\sum_{k=1}^{i+1}\frac{\|U_k\|_2}{\|W_k\|_2}\right) \overset{(iii)}{\leq} \\
&\overset{(iii)}{\leq} \left(\prod_{k=1}^{i+1}1+\eta_k\right)\left(\prod_{k=1}^{i+1}\|W_k\|_2\right)\left(\prod_{k=1}^{i}\mathrm{Lip}_k\right)\left(\|\Delta\mathbf{x}\|_2 + \|\mathbf{x}\|_2\sum_{k=1}^{i+1}\frac{\|U_k\|_2}{\|W_k\|_2}\right),
\end{aligned}
\tag{52}
$$

where (i) comes from Equation 51, (ii) comes from unrolling the recursion $\Delta_i$, (iii) comes from the fact that $1+\eta_k \geq 1$. $\qquad \square$

**Lemma 8** (GCN perturbation analysis). *Let $f_{\mathbf{w}} : \mathcal{X} \mapsto \mathbb{R}^K$ with $\mathbf{w} = vec\{W_1,...,W_n\}$ be a n-layer GCN model. Let $\psi_i$ be $\mathrm{Lip}_i$ activation functions after layer $i \in [n-1]$. If $\mathbf{u} = vec\{U_1,...,U_n\}$, $B \in \mathbb{R}$ are such that: $\forall i \in [n] : \|U_i\|_2 \leq \frac{1}{n}\|W_i\|_2$ and $\forall G \in \mathcal{X}, \forall v \in G : |z_v|_2 \leq B$, and $\forall G \in \mathcal{X}$, $G$ is simple and have maximum degree of $d$, then for any $G = (V,E,z) \in \mathcal{X}$*

$$
\begin{aligned}
|f_{\mathbf{w}+\mathbf{u}}(G) - f_{\mathbf{w}}(G)|_2 &\leq \frac{1}{\sqrt{|V|}}\|Z\|_F\|P_G\|_2^{n-1}\left(\prod_{i=1}^{n}\mathrm{Lip}_i\|W_i\|_2\right)\sum_{i=1}^{n}\frac{\|U_i\|_2}{\|W_i\|_2} \leq \\
&\leq eB\left(\prod_{i=1}^{n}\mathrm{Lip}_i\|W_i\|_2\right)\sum_{i=1}^{n}\frac{\|U_i\|_2}{\|W_i\|_2}, \leq \\
&\leq eBd^{(n-1)/2}\left(\prod_{i=1}^{n}\mathrm{Lip}_i\|W_i\|_2\right)\sum_{i=1}^{n}\frac{\|U_i\|_2}{\|W_i\|_2}
\end{aligned}
\tag{53}
$$

*where $Z \in \mathbb{R}^{|V|\times d_z}$ is the matrix consisting of $z_v$ for $v \in V$, and $P_G$ is a Laplacian of the graph $G$ which is equal to $\tilde{D}^{-1/2}\tilde{A}\tilde{D}^{-1/2}$ where $\tilde{A}$ is an adjacency matrix with $+1$ on the diagonal and $\tilde{D} = diag\left(\sum_{j=1}^{|V|}\tilde{A}_{ij}, i \in [|V|]\right)$ and $d$ is maximum degree of a graph in $\mathcal{X}$.*

*Proof.* The proof follows [48] except for the fact that we add $\mathrm{Lip}_i$ to the bound. This change is straightforward, however, for the completeness of the picture we provide this proof here. The detailed description of the architectures can be found in Appendix B.

First we prove two helpful propositions

**Proposition 1.** *For any matrix $A \in \mathbb{R}^{n\times m}, B \in \mathbb{R}^{m\times p}$, we have,*

$$
\|AB\|_F \leq \|A\|_F\|B\|_2
$$

*Proof.* Let $x_i^\top, a_i^\top$ be $i$th row of $AB$ and $A$ respectively, then we have:

$$
\|AB\|_F^2 \leq \sum_{i=1}^{n}\|x_i^\top|_2^2 = \sum_{i=1}^{n}\|a_i^\top B\|_2^2 \leq \sum_{i=1}^{n}\|a_i^\top\|_2^2\|B\|_2^2 = \|A\|_F^2\|B\|_2^2
$$

$\qquad \square$

**Proposition 2.** *For any undirected graph $G = (V,E)$, let $A \in \mathbb{R}^{|V|\times|V|}$ be the adjacency matrix, $D = \mathrm{diag}(D_1,\ldots,D_{|V|})$ be the degree matrix, and $d = \max_{i\in[|V|]}\{D_i\}$ be the maximum degree,*

where $D_i = \sum_{j=1}^{|V|} A_{ij}$, $i \in [|V|]$. *Then we have,*

(i) $\|A\|_2 \le d$;

(ii) $\left\|\tilde{D}^{-\frac{1}{2}}\tilde{A}\tilde{D}^{-\frac{1}{2}}\right\|_2 \le 1$, *where $\tilde{A} = A + I$ and $\tilde{D} = \mathrm{diag}(\tilde{D}_1, \ldots, \tilde{D}_{|V|}) = \mathrm{diag}\left(\sum_{j=1}^{|V|} \tilde{A}_{ij}, i \in [|V|]\right)$.*

*Proof.* For (i), by the definition of spectral norm, we have

$$\|A\|_2 = \max_{\|x\|_2=1} x^\top A x = \max_{\|x\|_2=1} \sum_{(i,j)\in E} x_i x_j \le \max_{\|x\|_2=1} \sum_{(i,j)\in E} \frac{x_i^2 + x_j^2}{2} \le \max_{\|x\|_2=1} d \sum_{i \in V} x_i^2 = d.$$

For (ii), let $\tilde{E} = E \cup \{(i,i) | i \in V\}$ be the edge set associated with the adjacency matrix $\tilde{A}$. By the definition of spectral norm, we have

$$\left\|D^{-\frac{1}{2}}\tilde{A}D^{-\frac{1}{2}}\right\|_2 = \max_{\|x\|_2=1} x^\top \left(D^{-\frac{1}{2}}\tilde{A}D^{-\frac{1}{2}}\right)x = \max_{\|x\|_2=1} \sum_{(i,j)\in\tilde{E}} \frac{x_i x_j}{\sqrt{\tilde{D}_i \tilde{D}_j}} \le$$

$$\le \max_{\|x\|_2=1} \sum_{(i,j)\in\tilde{E}} \left(\frac{x_i^2}{2\tilde{D}_i} + \frac{x_j^2}{2\tilde{D}_j}\right) \le \max_{\|x\|_2=1} \sum_{i \in V} x_i^2 = 1.$$

$\square$

Before proving the first inequality we note that $\frac{1}{\sqrt{|V|}}\|X\|_F \le B$ and $\|P_G\|_2^{n-1} \le 1$, so the second inequality is rather straightforward. The last inequality comes from the fact that $d \ge 1$.

We denote the node representation in the $j$-th ($j \le l$) layer as

$$f_\mathbf{w}^j(G) = H_j = \psi_j(P_G H_{j-1} W_j), \quad j \in [n-1],$$

$$f_\mathbf{w}^n(G) = H_n = \frac{1}{|V|}\mathbf{1}_{|V|} H_{n-1} W_n.$$

Adding perturbation $\mathbf{u}$ to the parameter $\mathbf{w}$, that is, for the $j$-th ($j \le n$) layer, the perturbed parameters are $W_j + U_j$ and denote $H_j' = f_{\mathbf{w}+\mathbf{u}}^j(G)$, $j \in [n]$.

**Upper Bound on the Node Representation.** For any $j < n$,

$$\|H_j\|_F = \|\psi_j(P_G H_{j-1} W_j)\|_F \le$$
$$\le \mathrm{Lip}_j\|P_G H_{j-1} W_j\|_F \le$$
$$\le \mathrm{Lip}_j\|P_G H_{j-1}\|_F\|W_j\|_2 \le$$
$$\le \mathrm{Lip}_j\|P_G\|_2\|H_{j-1}\|_F\|W_j\|_2,$$

where the first inequality holds since $\psi_j$ is a Lipschitz and $\psi_j(0) = 0$, and the second and the last ones hold by Proposition 1. Then, unrolling the recursion and setting $H_0 = X$, we have

$$\|H_j\|_F \le \|P_G\|_2^j\|H_0\|_F \prod_{i=1}^j \mathrm{Lip}_i\|W_i\|_2 \le$$

$$\le \|Z\|_F\|P_G\|_2^j \prod_{i=1}^j \mathrm{Lip}_i\|W_i\|_2. \tag{54}$$

**Upper Bound on the Change of Node Representation.** For any $j < |V|$,

$$\|H_j' - H_j\|_F = \|\psi_j(P_G H_{j-1}'(W_j + U_j)) - \psi_j(P_G H_{j-1} W_j)\|_F$$
$$\le \mathrm{Lip}_j\|P_G H_{j-1}'(W_j + U_j) - P_G H_{j-1} W_j\|_F \tag{55}$$

Using the triangle inequality, we have

$$\|H'_j - H_j\|_F \le \mathrm{Lip}_j \|P_G(W_j + U_j)(H'_{j-1} - H_{j-1})\|_F + \|P_G H_{j-1} U_j\|_F.$$

The first term can be bounded as

$$\|P_G(H'_{j-1} - H_{j-1})(W_j + U_j)\|_F = \|P_G\|_2 \|H'_{j-1} - H_{j-1}\|_F \|W_j + U_j\|_2,$$

and the second term can be bounded as

$$\|P_G H_{j-1} U_j\|_F = \|P_G\|_2 \|H_{j-1}\|_F \|U_j\|_2.$$

Therefore,

$$\|H'_j - H_j\|_F \le \mathrm{Lip}_j \|P_G\|_2 \|H'_{j-1} - H_{j-1}\|_F \|W_j + U_j\|_2 + \mathrm{Lip}_j \|P_G\|_2 \|H_{j-1}\|_F \|U_j\|_2. \quad (56)$$

Unrolling the recursion while simplifying notation as: $\|H_j - H'_j\|_F \le a_{j-1}\|H'_{j-1} - H_{j-1}\|_F + b_{j-1}$ we get:

$$\|H'_j - H_j\|_F \le \sum_{k=0}^{j-1} b_k \left( \prod_{i=k+1}^{j-1} a_i \right) =$$

$$= \sum_{k=0}^{j-1} \mathrm{Lip}_{k+1} \|P_G\| \|H_k\|_F \|U_{k+1}\|_2 \left( \prod_{i=k+1}^{j-1} \mathrm{Lip}_{i+1} \|P_G\|_2 \|W_{i+1} + U_{i+1}\|_2 \right) =$$

$$= \sum_{k=0}^{j-1} \|P_G\|_2^{j-k} \|H_k\|_F \|U_{k+1}\|_2 \prod_{i=k}^{j-1} \mathrm{Lip}_{i+1} \prod_{i=k+1}^{j-1} \|W_{i+1} + U_{i+1}\|_2.$$

Plugging in , we have:

$$\|H'_j - H_j\|_F \le \prod_{i=1}^{j} \mathrm{Lip}_i \sum_{k=0}^{j-1} \|P_G\|_2^{j-k} \left( \|P_G\|^k \|Z\|_F \prod_{i=1}^{k} \|W_i\|_2 \right) \|U_{k+1}\|_2 \prod_{i=k+1}^{j-1} \|W_i + U_i\|_2 \le$$

$$\le \|Z\|_F \prod_{i=1}^{j} \mathrm{Lip}_i \sum_{k=0}^{j-1} \|P_G\|_2^{j} \frac{\|U_{k+1}\|_2}{\|W_{k+1}\|_2} \prod_{i=1}^{k+1} \|W_i\|_2 \prod_{i=k+1}^{j-1} \left( 1 + \frac{1}{n} \right) \|W_i\|_2 =$$

$$= \|Z\|_F \|P_G\|_2^{j} \prod_{i=1}^{j} \|W_i\|_2 \prod_{i=1}^{j} \mathrm{Lip}_i \sum_{k=1}^{j} \frac{\|U_k\|_2}{\|W_k\|_2} \left( 1 + \frac{1}{n} \right)^{j-k}$$

$$(57)$$

**Final Bound on the Readout Layer.**

$$\|H'_n - H_n\|_2 = \left\| \frac{1}{|V|} \mathbf{1}_{|V|} H'_{n-1}(W_n + U_n) - \frac{1}{|V|} \mathbf{1}_{|V|} H_{n-1} W_n \right\|_2 =$$

$$= \left\| \frac{1}{|V|} \mathbf{1}_{|V|}(H'_{n-1} - H_{n-1})(W_n + U_n) + \frac{1}{|V|} \mathbf{1}_{|V|} H_{n-1} U_n \right\|_2 \le$$

$$\le \left\| \frac{1}{|V|} \mathbf{1}_{|V|}(H'_{n-1} - H_{n-1})(W_n + U_n) \right\|_2 + \left\| \frac{1}{|V|} \mathbf{1}_{|V|} H_{n-1} U_n \right\|_2 \le$$

$$\le \left\| \frac{1}{|V|} \mathbf{1}_{|V|} \right\|_2 \|(H'_{n-1} - H_{n-1})(W_n + U_n)\|_2 + \left\| \frac{1}{|V|} \mathbf{1}_{|V|} \right\|_2 \|H_{n-1} U_n\|_2 \le$$

$$\le \frac{1}{\sqrt{|V|}} \|H'_{n-1} - H_{n-1}\|_F \|W_n + U_n\|_2 + \frac{1}{\sqrt{|V|}} \|H_{n-1}\|_F \|U_n\|_2,$$

where in a last inequality we first apply that $\|A\|_2 \le \|A\|_F$ for any matrix $A$ and then use Proposition 1.

Using  and , we have:

Figure 5: Case $g(n(k)) > g(m(k)), f(n(k)) \geq f(m(k))$

Figure 6: Case $g(n(k)) > g(m(k)), f(n(k)) \leq f(m(k))$

$$\|H'_n - H_n\|_2 \leq \frac{1}{\sqrt{|V|}} \|W_n + U_n\|_2 \|X\|_F \|P_G\|^{n-1} \prod_{i=1}^{n-1} \mathrm{Lip}_i \|W_i\|_2 \sum_{k=1}^{n-1} \frac{\|U_k\|_2}{\|W_k\|_2} \left(1 + \frac{1}{n}\right)^{n-k-1} +$$

$$+ \frac{1}{\sqrt{|V|}} \|U_n\|_2 \|P_G\|_2^{n-1} \|X\|_F \prod_{i=1}^{n-1} \mathrm{Lip}_i \|W_i\|_2 \leq$$

$$\leq \frac{1}{\sqrt{|V|}} \|Z\|_F \|P_G\|^{n-1} \left[ \|W_n + U_n\|_2 \prod_{i=1}^{n-1} \mathrm{Lip}_i \|W_i\|_2 \sum_{k=1}^{n-1} \frac{\|U_k\|_2}{\|W_k\|_2} \left(1 + \frac{1}{n}\right)^{n-k-1} + \|U_n\|_2 \prod_{i=1}^{n-1} \mathrm{Lip}_i \|W_i\|_2 \right]$$

$$= \frac{1}{\sqrt{|V|}} \|Z\|_F \|P_G\|^{n-1} \prod_{i=1}^{n} \mathrm{Lip}_i \|W_i\|_2 \left[ \frac{\|W_n + U_n\|_2}{\|W_n\|_2} \sum_{k=1}^{n-1} \frac{\|U_k\|_2}{\|W_k\|_2} \left(1 + \frac{1}{n}\right)^{n-k-1} + \frac{\|U_n\|_2}{\|W_n\|_2} \right] \leq$$

$$\leq \frac{1}{\sqrt{|V|}} \|Z\|_F \|P_G\|_2^{n-1} \prod_{i=1}^{n} \mathrm{Lip}_i \|W_i\|_2 \left[ \left(1 + \frac{1}{n}\right) \sum_{k=1}^{n-1} \frac{\|U_k\|_2}{\|W_k\|_2} \left(1 + \frac{1}{n}\right)^{n-k-1} + \frac{\|U_n\|_2}{\|W_n\|_2} \right] \leq$$

$$\leq \frac{e}{\sqrt{|V|}} \|Z\|_F \|P_G\|_2^{n-1} \prod_{i=1}^{n} \mathrm{Lip}_i \|W_i\|_2 \sum_{k=1}^{n} \frac{\|U_k\|_2}{\|W_k\|_2},$$

where we set $\mathrm{Lip}_n = 1$ for simplicity of notation and last inequality holds since $1 \leq (1 + \frac{1}{n})^n \leq e$.

$\square$

Our next lemma helps us to upper bound the PERSLAY's perturbation, when $\mathrm{Agg} = k\text{-}\max$.

**Lemma 9.** *Let $X$ be an arbitrary finite set and $f, g : X \mapsto \mathbb{R}$. Then we can say that:*

$$|k\text{-}\max_{x \in X} f(x) - k\text{-}\max_{x \in X} g(x)| \leq 3 \max_{x \in X} |f(x) - g(x)|$$

*Proof.* Denote $n : \mathbb{N} \mapsto X$ by a function that maps natural number $k$ to an element of $X$ that would be on $k$th position in order sorted by $f$. Denote $m$ as an analogous function but for $g$. Then, we are interested in the following expression: $|f(n(k)) - g(m(k))|$. Let us rewrite it:

$$|f(n(k)) - g(m(k))| = |f(n(k)) - g(n(k)) + g(n(k)) - g(m(k))| \leq$$
$$\leq |f(n(k)) - g(n(k))| + |g(n(k)) - g(m(k))| \leq$$
$$\leq \max_{x \in X} |f(x) - g(x)| + |g(n(k)) - g(m(k))|$$

Now, the task is to prove that $|g(n(k)) - g(m(k))| \leq 2 \max_{x \in X} |f(x) - g(x)|$

Let us consider four cases:

- $g(n(k)) > g(m(k))$ and $f(n(k)) \geq f(m(k))$ (Fig. 5). In this case $\exists i \in \mathbb{N}$ such that $f(n(i)) > f(n(k))$ and $g(n(i)) < g(m(k))$. Indeed, if none of the elements "to the right"

of $n(k)$ moved "to the left" of $m(k)$, then "to the right" of $m(k)$, there are at least $n - k + 1$ elements; however, there are must be exactly $n - k$ elements.

$$|g(n(k)) - g(m(k))| = g(n(k)) - g(m(k)) \leq g(n(k)) - g(n(i)) \leq$$
$$\leq f(n(k)) + (\max_{x \in X} |f(x) - g(x)|) - g(n(i)) <$$
$$< f(n(i)) + (\max_{x \in X} |f(x) - g(x)|) - g(n(i)) < 2(\max_{x \in X} |f(x) - g(x)|)$$

- $g(n(k)) > g(m(k))$ and $f(n(k)) \leq f(m(k))$ (Fig. 6)

$$|g(n(k)) - g(m(k))| = g(n(k)) - g(m(k)) \leq$$
$$\leq f(n(k)) + \left( \max_{x \in X} |f(x) - g(x)| \right) - g(m(k)) \leq$$
$$\leq f(m(k)) + \left( \max_{x \in X} |f(x) - g(x)| \right) - g(m(k)) \leq$$
$$\leq 2 \left( \max_{x \in X} |f(x) - g(x)| \right)$$

- The rest of the cases can be handled analogously.

$\square$

**Lemma 10** (Perturbation analysis of PersLay). *Let $f_{\mathbf{w}} : \mathcal{G} \mapsto \mathbb{R}^k$ with $\mathbf{w} = \{W^{(\omega)}, W^{(\varphi)}\}$ be a PersLay where $W^{(\omega)}$ is a parameter vector (matrix) of weight function, $\omega$, and $W^{(\varphi)}$ is a parameter vector (matrix) of point-transformation function, $\varphi$. Let $\mathrm{Dg}$ be a mapping from graphs to (extended) persistence diagrams with a fixed filtration function and $B$ such that $\max_{G \in \mathcal{G}} \max_{p \in \mathrm{Dg}(G)} \|p\|_2 \leq B$, then:*

$$\max_{G \in \mathcal{G}} |f_{\mathbf{w}}(G)|_2 \leq A_1 C^{(\omega)} T^{(\omega)} C^{(\varphi)} T^{(\varphi)} \tag{58}$$

*and for $\eta^{(\omega)}$ and $\mathbf{u} = \{U^{(\omega)}, U^{(\varphi)}\}$ such that $\|U^{(\omega)}\|_2 \leq \eta^{(\omega)} T^{(\omega)}$, we have:*

$$\max_{G \in \mathcal{G}} |f_{\mathbf{w}+\mathbf{u}}(G) - f_{\mathbf{w}}(G)|_2 \leq$$
$$\leq A_2 \max\{C^{(\omega)} Lip^{(\varphi)}, C^{(\varphi)} Lip^{(\omega)}\}(1 + \eta^{(\omega)}) T^{(\omega)} T^{(\varphi)} \left( \frac{\|U^{(\varphi)}\|_2}{T^{(\varphi)}} + \frac{\|U^{(\omega)}\|_2}{T^{(\omega)}} \right), \tag{59}$$

*where $A_1 = \max_{G \in \mathcal{G}} card(\mathrm{Dg}(G))$ if Agg is sum and $A_1 = 1$ if Agg is k-max or mean; $A_2 = \max_{G \in \mathcal{G}} card(\mathrm{Dg}(G))$ if Agg is sum or $A_2 = 3$ if Agg is k-max or mean;*

$$(T^{(\varphi)}, C^{(\varphi)}, Lip^{(\varphi)}) = (\max\{1, \|W^{(\varphi)}\|_2\}, \sqrt{h}B, 1) \qquad \varphi = \Lambda$$
$$(T^{(\varphi)}, C^{(\varphi)}, Lip^{(\varphi)}) = (\max\{1, \|W^{(\varphi)}\|_2\}, \sqrt{h}, \tau e^{-1/2}) \qquad \varphi = \Gamma$$
$$(T^{(\varphi)}, C^{(\varphi)}, Lip^{(\varphi)}) = (\|W^{(\varphi)}\|_2, \sqrt{3}\max\{1, B\}, \max\{1, B\}) \qquad \varphi = L$$

$T^{(\omega)} = \max\{1, \|W^{(\omega)}\|_2\}$; *and $C^{(\omega)}, Lip^{(\omega)}$ are such that:*

$$\max_{G \in \mathcal{G}} \max_{p \in \mathrm{Dg}(G)} |\omega_{\mathbf{w}}(p)|_2 \leq C^{(\omega)} T^{(\omega)} \quad and \quad \max_{G \in \mathcal{G}} \max_{p \in \mathrm{Dg}(G)} |\omega_{\mathbf{w}+\mathbf{u}}(p) - \omega_{\mathbf{w}}(p)| \leq Lip^{(\omega)} \|U^{(\omega)}\|_2$$

*for any $\mathbf{w}$ and $\mathbf{u}$.*

*Proof.* First we prove the inequaliry about maximum output norm.

**Maximum output norm.**

$$\max_{G\in\mathcal{G}}\|f(\mathrm{Dg}(G))\|_2 \leq A_1\max_{G\in\mathcal{G}}\max_{p\in\mathrm{Dg}(G)}\|\omega_{\mathbf{w}}(p)\varphi_{\mathbf{w}}(p)\|_2 \leq$$

$$\leq A_1\max_{G\in\mathcal{G}}\max_{p\in\mathrm{Dg}(G)}\omega_{\mathbf{w}}(p)\|\varphi_{\mathbf{w}}(p)\|_2 \leq \qquad(60)$$

$$\leq A_1\max_{G\in\mathcal{G},p\in\mathrm{Dg}(G)}\omega_{\mathbf{w}}(p)\max_{G\in\mathcal{G},p\in\mathrm{Dg}\,G}\|\varphi_{\mathbf{w}}(p)\|_2,$$

where $A_1$ is $\max\limits_{G\in\mathcal{G}} card(\mathrm{Dg}(G))$ if Agg is sum and $A_1 = 1$ if Agg is $k$-max or mean.

From the Lemma statement we can upper bound $\omega_{\mathbf{w}}(p)$:

$$\max_{G\in\mathcal{G},p\in\mathrm{Dg}(G)}\omega_{\mathbf{w}}(p) \leq C^{(\omega)}T^{(\omega)}$$

**Maximum norm of $\varphi$:**

$\varphi = \Lambda$.
$$\max_{G\in\mathcal{G},p\in\mathrm{Dg}(G)}|\varphi_{\mathbf{w}}(p)_i| = \max_{G\in\mathcal{G},p\in\mathrm{Dg}(G)}\max\{0, p_2 - |t_i - p_1|\} \leq p_2 \leq B$$

Hence:

$$\max_{G\in\mathcal{G},p\in\mathrm{Dg}(G)}\|\varphi_{\mathbf{w}}(p)\|_2 = \left[\sum_{i=1}^{h}\varphi_{\mathbf{w}}(p)_i^2\right]^{1/2} \leq \left[\sum_{i=1}^{h}b^2\right]^{1/2} = B\sqrt{h}$$

$\varphi = \Gamma$.
$$\max_{G\in\mathcal{G},p\in\mathrm{Dg}(G)}|\varphi_{\mathbf{w}}(p)_i| = \max_{G\in\mathcal{G},p\in\mathrm{Dg}(G)}\exp\left[-\frac{|p_1 - t_{i,1}|^2 + |p_2 - t_{i,2}|^2}{2\tau^2}\right] \leq 1$$

Hence:

$$\max_{G\in\mathcal{G},p\in\mathrm{Dg}(G)}\|\varphi_{\mathbf{w}}(p)\|_2 = \left[\sum_{i=1}^{h}|\varphi_{\mathbf{w}}(p)_i|^2\right]^{1/2} \leq \left[\sum_{i=1}^{h}1\right]^{1/2} = \sqrt{h}$$

$\varphi = \Psi$.
$$\max_{G\in\mathcal{G},p\in\mathrm{Dg}(G)}|\varphi_{\mathbf{w}}(p)_i| = \max_{G\in\mathcal{G},p\in\mathrm{Dg}(G)}|p_1 t_i[1] + p_2 t_i[2] + t_i[3]| \leq$$

$$\leq \max\{B,1\}|t_{i,1} + t_{i,2} + t_i[3]|$$

Hence:

$$\max_{G\in\mathcal{G},p\in\mathrm{Dg}(G)}\|\varphi_{\mathbf{w}}(p)\|_2 = \left[\sum_{i=1}^{h}|(\varphi_{\mathbf{w}})_i|^2\right]^{1/2} \leq \left[\sum_{i=1}^{h}\max\{B,1\}^2|t_{i,1} + t_{i,2} + t_i[3]|^2\right]^{1/2} \leq$$

$$\leq \max\{b,1\}\left[\sum_{i=1}^{h}3(t_{i,1}^2 + t_{i,2}^2 + t_i[3]^2)\right] \leq$$

$$\leq \sqrt{3}\max\{B,1\}\|vec(W^{(\varphi)})\|_2$$

Combining all together we get the Equation 58.

**Maximum perturbation of the PersLay output.**

$$\max_{G\in\mathcal{G}}\|f_{\mathbf{w}+\mathbf{u}}(\mathrm{Dg}(G)) - f_{\mathbf{w}}(\mathrm{Dg}(G))\|_2 \leq A_2\max_{G\in\mathcal{G},p\in\mathrm{Dg}(G)}\|\omega_{\mathbf{w}+\mathbf{u}}(p)\varphi_{\mathbf{w}+\mathbf{u}}(p) - \omega_{\mathbf{w}}(p)\varphi_{\mathbf{w}}(p)\|_2 \leq$$

$$\leq A_2\max_{G\in\mathcal{G},p\in\mathrm{Dg}(G)}[\|\omega_{\mathbf{w}+\mathbf{u}}(p)(\varphi_{\mathbf{w}+\mathbf{u}}(p) - \varphi_{\mathbf{w}}(p)) + \varphi_{\mathbf{w}}(p)(\omega_{\mathbf{w}+\mathbf{u}}(p) - \omega_{\mathbf{w}}(p))\|_2] \leq$$

$$\leq A_2\max_{G\in\mathcal{G},p\in\mathrm{Dg}(G)}[\omega_{\mathbf{w}+\mathbf{u}}(p)|\|\varphi_{\mathbf{w}+\mathbf{u}}(p) - \varphi_{\mathbf{w}}(p)\|_2 + \|\varphi_{\mathbf{w}}(p)\|_2|\omega_{\mathbf{w}+\mathbf{u}}(p) - \omega_{\mathbf{w}}(p)|],$$

where $A_2$ is $\max\limits_{G \in \mathcal{G}} card(\mathrm{Dg}(G))$ if Agg is sum, $A_2$ is 3 if Agg is $k$-max by [Lemma 9] and $A_2$ is 1 if Agg is mean.

By the Lemma statement we have:

$$\max_{G \in \mathcal{G}, p \in \mathrm{Dg}(G)} \|\omega_{\mathbf{w}+\mathbf{u}}\|_2 \leq C^\omega (1 + \eta^{(\omega)}) T^{(\omega)},$$

and

$$\max_{G \in \mathcal{G}, p \in \mathrm{Dg}(G)} |\omega_{\mathbf{w}+\mathbf{u}}(p) - \omega_{\mathbf{w}}(p)| \leq \mathrm{Lip}^{(\omega)} \|U^{(\omega)}\|_2.$$

Moreover, from [Paragraph about max norm of $\varphi$]:

$$\max_{G \in \mathcal{G}, p \in \mathrm{Dg}(G)} \|\varphi_{\mathbf{w}}(p)\|_2 \leq C^{(\varphi)} T^{(\varphi)}$$

**Maximum perturbation of $\varphi$:**

$\varphi = \Lambda$. Since $g(x) = |x|$ is 1-Lipschitz we have that:

$$\max_{G \in \mathcal{G}, p \in \mathrm{Dg}(G)} \|(\varphi_{\mathbf{w}+\mathbf{u}}(p) - \varphi_{\mathbf{w}}(p))_i\| \leq U_i^{(\varphi)}$$

Hence,

$$\max_{G \in \mathcal{G}, p \in \mathrm{Dg}(G)} \|\varphi_{\mathbf{w}+\mathbf{u}}(p) - \varphi_{\mathbf{w}}(p)\|_2 \leq \left[ \sum_{i=1}^h |(\varphi_{\mathbf{w}+\mathbf{u}}(p) - \varphi_{\mathbf{w}}(p))_i|^2 \right]^{1/2} \leq$$

$$\leq \left[ \sum_{i=1}^h (U_i^{(\varphi)})^2 \right]^{1/2} = \|U^{(\varphi)}\|_2$$

$\varphi = \Gamma$. Suppose $g(x, y) = \exp\left(-\frac{x^2 + y^2}{2\tau^2}\right)$. Then

$$\|g(x, y) - g(x + \Delta x, y + \Delta y)\|_2 = |\nabla g(x', y')|_2 \sqrt{\Delta x^2 + \Delta y^2} \leq$$
$$\leq \max_{x', y'} |\nabla g(x', y')|_2 \sqrt{\Delta x^2 + \Delta y^2}$$

by mean-value theorem for some $x', y'$ between $x$ and $x + \Delta x$ and $y$ and $y + \Delta y$.

Let us find maximum of the gradient by every coordinate. Since the function is symmetric we need to do it only for one of the coordinates.

The maximum of the norm of the first coordinate of the gradient is achieving at the point $t_{i,1} = p_1 \pm \tau$, and the gradient value at these points is at most $\frac{1}{\tau e^{1/2}}$. So we have:

$$\max_{G \in \mathcal{G}, p \in \mathrm{Dg}(G)} \|\varphi_{\mathbf{w}+\mathbf{u}}(p) - \varphi_{\mathbf{w}}(p)\|_2 \leq \left[ \sum_{i=1}^h \frac{\|U_i^{(\varphi)}\|_2^2}{\tau^2 e} \right]^{1/2} \leq \frac{\|vec(U^{(\varphi)})\|_2}{\tau e^{1/2}}$$

$\varphi = \Psi$. In this case $g(x, y, z) = p_1 x + p_2 y + z$ is $\max\{B, 1\}$-Lipschitz, so

$$\max_{G \in \mathcal{G}, p \in \mathrm{Dg}(G)} \|\varphi_{\mathbf{w}+\mathbf{u}}(p) - \varphi_{\mathbf{w}}(p)\|_2 \leq \left[ \sum_{i=1}^h \max\{B, 1\}^2 \|U_i^{(\varphi)}\|_2^2 \right]^{1/2} = \max\{B, 1\} \|vec(U^{(\varphi)})\|_2^2$$

Combining all together we have:

$$\max_{G \in \mathcal{G}} \|f_{\mathbf{w}+\mathbf{u}}(G) - f_{\mathbf{w}}(G)\|_2 \leq A_2 \left[ (1 + \eta^{(\omega)}) C^{(\omega)} T^{(\omega)} \mathrm{Lip}^{(\varphi)} \|U^{(\varphi)}\|_2 + C^{(\varphi)} T^{(\varphi)} \mathrm{Lip}^{(\omega)} \|U^{(\omega)}\|_2 \right] \leq$$

$$\leq A_2 T^{(\varphi)} T^{(\omega)} (1 + \eta^{(\varphi)}) \max\{\mathrm{Lip}^{(\varphi)} C^{(\omega)}, \mathrm{Lip}^{(\omega)} C^{(\varphi)}\} \left( \frac{\|U^{(\varphi)}\|_2}{\|T^{(\varphi)}\|_2} + \frac{\|U^{(\omega)}\|_2}{\|T^{(\omega)}\|_2} \right)$$

$$\square$$

# E Proofs of corollaries

**Corollary 3.** *Let $f_{\mathbf{w}} : \mathcal{X} \mapsto \mathbb{R}^K$ with $\mathbf{w} = \{W_1, ..., W_n\}$ be an $n$-layer MLP with $Lip_i$-Lipschitz activation functions $\psi_i$, for $i \in [n-1]$. Let $B \in \mathbb{R}$ be such that $\forall x \in \mathcal{X} : \|x\|_2 \leq B$. Then $f_{\mathbf{w}}$ satisfy requirements of Lemma 2 with:*

$$T = \prod_{i=1}^{n} \|W_i\|_2 \quad and \quad \forall i \in [n] : \ S_i = \|W_i\|_2, \eta_i = \frac{1}{6n}$$

$$C_1 = B \prod_{i=1}^{n-1} Lip_i \quad and \quad C_2 = eB \prod_{i=1}^{n-1} Lip_i$$

*Proof.* To test Equation 6 and Equation 7 we use Lemma 7.

To check Equation 8 we apply arithmetic-geometric mean inequality:

$$\prod_{i=1}^{n} \|W_i\|_2 \left( \sum_{i=1}^{n} \frac{1}{\|W_i\|_2} \right) = \sum_{i=1}^{n} \prod_{j \neq i} \|W_j\|_2 \geq$$
$$\geq n \left( \prod_{i=1}^{n} \prod_{j \neq i} \|W_j\|_2 \right) = n \left( \prod_{i=1}^{n} T_i \right)^{\frac{n-1}{n}} = nT^{\frac{n-1}{n}}$$

(61)

It is left to show that Equation 9 holds:

$$\frac{C_1}{2C_2 \frac{1}{6n}} \geq \frac{3n}{e} \geq 1$$

(62)

$\square$

**Corollary 4.** *Let $f_{\mathbf{w}} : \mathcal{X} \mapsto \mathbb{R}^k$ with $\mathbf{w} = \{W_1, ..., W_n\}$ be a $n$-layer GCN network with readout layer. Let $\psi_i$ for $i \in [n-1]$ be a $Lip_i$-Lipschitz activation function. Let node feature of any graph be contained in $\ell_2$-ball of radius $B$, i.e. $\forall G \in \mathcal{X} : \|z_v\|_2 \leq B$ for every node $v$ and $\forall G \in \mathcal{X}$, $G$ is simple and has maximum degree at most $d-1$. Then $f_{\mathbf{w}}$ satisfy requirements of Lemma 2 with:*

$$T = \prod_{i=1}^{n} \|W_i\|_2 \quad and \quad \forall i \in n : \ S_i = \|W_i\|_2, \ \eta_i = \frac{1}{6n}$$

*if using perturbation analysis from Liao et al. [33]*

$$C_1 = d^{\frac{n-1}{2}} B \prod_{i=1}^{n-1} Lip_i \quad and \quad C_2 = eBd^{\frac{n-1}{2}} \prod_{i=1}^{n-1} Lip_i$$

*or if using perturbation analysis from Sun and Lin [48]*

$$C_1 = B \prod_{i=1}^{n-1} Lip_i \quad and \quad C_2 = eB \prod_{i=1}^{n-1} Lip_i$$

*Proof.* To test Equation 6, Equation 7 we use Lemma 8. We can apply Lemma 8 because $\bar{\eta} = \frac{1}{6n} \leq \frac{1}{n}$.

To test Equation 8 we use arithmetic vs geometric mean inequality:

$$\prod_{i=1}^{n} \|W_i\|_2 \left( \sum_{i=1}^{n} \frac{1}{\|W_i\|_2} \right) = \sum_{i=1}^{n} \prod_{j \neq i} \|W_j\|_2 \geq$$
$$\geq n \left( \prod_{i=1}^{n} \prod_{j \neq i} \|W_j\|_2 \right) = n \left( \prod_{i=1}^{n} T_i \right)^{\frac{n-1}{n}} = nT^{\frac{n-1}{n}}$$

It is left to test Equation 9

$$\frac{C_1}{2C_2 \frac{1}{6n}} \geq \frac{3n}{e} \geq 1$$

$\square$

**Corollary 5.** *Let $f_{\mathbf{w}} : \mathcal{X} \mapsto \mathbb{R}^k$ with $\mathbf{w} = \{W_1, W_2, W_3\}$ be $n$-layer MPGNN ($n > 2$). Let $g, \phi, \rho$ be activations functions with Lipschitz constants: $Lip_g, Lip_\phi, Lip_\rho$. We denote $Lip_g Lip_\phi Lip_\rho \|W_2\|_2$ with $\mathcal{C}$. Let node feature of any graph be contained in $\ell_2$-ball of radius $B$, i.e. $\forall G = (V, E, z) \in \mathcal{X}, \forall v \in V : \|z_v\|_2 \leq B$ and $\forall G \in \mathcal{X}$, $G$ is simple and has maximum degree at most $d - 1$. Then $f_{\mathbf{w}}$ satisfy Lemma 2 requirements with*

*if $d\mathcal{C} \neq 1$:*

$$T = \|W_1\|_2 \|W_3\|_2 \frac{(d\mathcal{C})^{n-1} - 1}{d\mathcal{C} - 1},$$

$$S_1 = \|W_1\|, \ S_2 = \min\{d\mathcal{C}, \|W_2\|_2\}, \ S_3 = \|W_3\|_2,$$

$$\eta_1 = \eta_2 = \eta_3 = \frac{1}{6n}$$

$$C_1 = BC_\phi \quad and \quad C_2 = eBC_\phi n$$

*if $d\mathcal{C} = 1$:*

$$T = \|W_1\|_2 \|W_3\|_2$$

$$S_1 = \|W_1\|_2, \ S_2 = \min\{1, \|W_2\|_2\}, \ S_3 = \|W_3\|_2$$

$$\eta_1 = \eta_2 = \eta_3 = \frac{1}{6(n+1)}$$

$$C_1 = BC_\phi(n+1) \quad and \quad C_2 = eBC_\phi(n+1)^2$$

*Proof.* From [33] (Lemma 3.3) we know that:

$$\max_{G \in \mathcal{X}} \|f_{\mathbf{w}+\mathbf{u}}(G) - f_{\mathbf{w}}(G)\|_2 \leq \begin{cases} eB(n+1)^2 \eta \|W_1\|_2 \|W_3\|_2 C_\phi, & d\mathcal{C} = 1 \\ eBn\eta \|W_1\|_2 \|W_3\|_2 C_\phi \frac{(d\mathcal{C})^{n-1}-1}{d\mathcal{C}-1}, & d\mathcal{C} \neq 1, \end{cases} \tag{63}$$

where $\eta = \max\left\{\frac{\|U_1\|_2}{\|W_1\|_2}, \frac{\|U_2\|_2}{\|W_2\|_2}, \frac{\|U_3\|_2}{\|W_3\|_2}\right\} \leq \frac{1}{n}$. In our case $\bar{\eta} \leq \frac{1}{6n}$, so we can apply these inequalities. Note that in our notation we have $n$-layer MPGNN and instead of $W_l$ we have $W_3$ (and instead of $U_l$ we have $U_3$).

And

$$\max_{G \in \mathcal{X}} \|f_{\mathbf{w}}(G)\|_2 \leq \begin{cases} B(n-1)C_\phi \|W_1\|_2 \|W_3\|_2, & d\mathcal{C} = 1 \\ BC_\phi \|W_1\|_2 \|W_3\|_2 \frac{(d\mathcal{C}^{n-1})-1}{d\mathcal{C}-1}, & d\mathcal{C} \neq 1 \end{cases} \tag{64}$$

Eq. (76) and Eq. (68) respectively.

Equation 6 follows from Equation 64.

To test Equation 7 we note that:

$$\eta \leq \frac{\|U_1\|_2}{\|W_1\|_2} + \frac{\|U_2\|_2}{\|W_2\|_2} + \frac{\|U_3\|_2}{\|W_3\|_2} \leq \frac{\|U_1\|_2}{\|W_1\|_2} + \frac{\|U_2\|_2}{\min\{d\mathcal{C}, \|W_2\|_2\}} + \frac{\|U_3\|_2}{\|W_3\|_2} \tag{65}$$

Now Equation 7 follows from Equation 63 and Equation 65.

To test Equation 8 we employ arithmetic vs geometric mean inequality:

$$T\left(\frac{1}{S_1} + \frac{1}{S_2} + \frac{1}{S_3}\right) \geq \frac{3T}{(S_1 S_2 S_3)^{1/3}} \geq \begin{cases} 3\frac{\|W_1\|_2 \|W_3\|_2}{(\|W_1\|_2 \|W_3\|_2)^{1/3}} \geq 3T^{2/3}, & d\mathcal{C} = 1 \\ 3\frac{\|W_1\|_2 \|W_3\|_2 \frac{(d\mathcal{C})^{n-1}-1}{d\mathcal{C}-1}}{(\|W_1\|_2 d\mathcal{C} \|W_3\|_2)^{1/3}} \geq 3T^{2/3}, & d\mathcal{C} \neq 1 \end{cases},$$

where in case $d\mathcal{C} = 1$, $S_2 \leq 1$ and in case $d\mathcal{C} \neq 1$, $S_2 \leq d\mathcal{C}$ and $\frac{(d\mathcal{C})^{n-1}-1}{d\mathcal{C}-1} \geq d\mathcal{C}$ for $n > 2$.

To test Equation 9 we provide the following derivation:

$$\frac{C_1}{2C_2} \geq \begin{cases} \frac{l+1}{2e(n+1)^2} \geq \frac{1}{6(n+1)}, & d\mathcal{C} = 1 \\ \frac{1}{en} \geq \frac{1}{6n}, & d\mathcal{C} \neq 1 \end{cases}$$

$\square$

**Corollary 1.** *Let* $f_\mathbf{w} : \mathcal{G} \mapsto \mathbb{R}^k$ *with* $\mathbf{w} = \{W^{(\omega)}, W^{(\varphi)}\}$ *be a PersLay where* $W^{(\omega)}$ *is a parameter vector (matrix) of weight function,* $\omega$*, and* $W^{(\varphi)}$ *is a parameter vector (matrix) of point-transformation function,* $\varphi$*. Let* Dg *be a mapping from graphs to (extended) persistence diagrams with a fixed filtration function and B such that* $\max\limits_{G \in \mathcal{G}} \max\limits_{p \in \mathrm{Dg}(G)} \|p\|_2 \leq B$*, then* $f_\mathbf{w}$ *satisfy requirements of Lemma 2 with:*

$$T = T^{(\varphi)}T^{(\omega)} \quad and \quad (S^{(\varphi)}, S^{(\omega)}) = (\|W^{(\varphi)}\|_2, \|W^{(\omega)}\|_2) \quad and \quad (\eta^{(\varphi)}, \eta^{(\omega)}) = (1,1)$$

$$C_1 = 2 \max \left\{ A_1 C^{(\omega)} C^{(\varphi)}, 2A_2 \max\{C^{(\omega)} Lip^{(\varphi)}, C^{(\varphi)} Lip^{(\omega)}\} \right\}$$

$$C_2 = 2A_2 \max\{C^{(\omega)} Lip^{(\varphi)}, C^{(\varphi)} Lip^{(\omega)}\},$$

*where* $A_1 = \max\limits_{G \in \mathcal{G}} card(\mathrm{Dg}(G))$ *if Agg is sum and* $A_1 = 1$ *if Agg is k-max or mean;* $A_2 = \max\limits_{G \in \mathcal{G}} card(\mathrm{Dg}(G))$ *if Agg is sum or* $A_2 = 3$ *if Agg is k-max or mean;*

$$(T^{(\varphi)}, C^{(\varphi)}, Lip^{(\varphi)}) = (\max\{1, \|W^{(\varphi)}\|_2\}, B\sqrt{h}, 1) \qquad \varphi = \Lambda$$

$$(T^{(\varphi)}, C^{(\varphi)}, Lip^{(\varphi)}) = (\max\{1, \|W^{(\varphi)}\|_2\}, \sqrt{h}, \tau e^{-1/2}) \qquad \varphi = \Gamma$$

$$(T^{(\varphi)}, C^{(\varphi)}, Lip^{(\varphi)}) = (\|W^{(\varphi)}\|_2, \sqrt{3}\max\{1, B\}, \max\{1, B\}) \qquad \varphi = \Psi$$

$T^{(\omega)} = \max\{1, \|W^{(\omega)}\|_2\}$*; and* $C^{(\omega)}, Lip^{(\omega)}$ *are such that:*

$$\max\limits_{G \in \mathcal{G}} \max\limits_{p \in \mathrm{Dg}(G)} |\omega_\mathbf{w}(p)|_2 \leq C^{(\omega)}T^{(\omega)} \quad and \quad \max\limits_{G \in \mathcal{G}} \max\limits_{p \in \mathrm{Dg}(G)} |\omega_{\mathbf{w+u}}(p) - \omega_\mathbf{w}(p)| \leq Lip^{(\omega)}\|U^{(\omega)}\|_2$$

*for any* $\mathbf{w}$ *and* $\mathbf{u}$*.*

*Proof.* To test Equation 6 and Equation 7 we apply Lemma 10. $\eta^{(\omega)}$ in Lemma 10 is arbitrary, so we can use this lemma with $\eta^{(\omega)} = 1$ and in this case $1 + \eta^{(\omega)} = 2$, so our choice of $C_2$ works.

To test Equation 8 we use arithmetic vs geometric mean inequality:

$$T^{(\varphi)}T^{(\omega)} \left( \frac{1}{S^{(\varphi)}} + \frac{1}{S^{(\omega)}} \right) \geq 2\frac{T^{(\varphi)}T^{(\omega)}}{(\|W^{(\omega)}\|_2 \|W^{(\varphi)}\|_2)^{1/3}} \geq 2\frac{T^{(\omega)}T^{(\varphi)}}{(T^{(\omega)}T^{(\varphi)})^{1/3}} = 2T^{2/3}$$

It is left to test Equation 9

$$\frac{C_1}{2C_2} \geq \frac{2A_1 C^{(\omega)} C^{(\varphi)} \max\left\{1, \frac{2A_2 \max\{C^{(\omega)}\mathrm{Lip}^{(\varphi)}, C^{(\varphi)}\mathrm{Lip}^{(\omega)}\}}{A_1 C^{(\omega)} C^{(\varphi)}}\right\}}{2A_2 \max\{C^{(\omega)}\mathrm{Lip}^{(\varphi)}, C^{(\varphi)}\mathrm{Lip}^{(\omega)}} \geq 1$$

$\square$

# F  Comparing bounds with prior works

**MLP & GCN**  The result is not as tight as in [40] because of the homogenity assumption that they do. Specificaly, matching our result to theirs, we have $B^2(h \ln nh)\bar{\eta}^{-2}$ the same as $B^2 n^2 (h \ln nh)$ (instead of $n$ they have $d$ in their notation). We have $\log\left(\frac{m}{\delta}\max\{1, \frac{1}{B}\}\right)$ which is in most cases better than $\log\frac{m \cdot n}{\delta}$. However, the suboptimality of our bound comes from:

$$\sum_{i=1}^n \frac{\|W_i\|_F^2}{\|W_i\|_2^2} \leq \sum_{i=1}^n \|W_i\|_F^2 \sum_{i=1}^n \frac{1}{\|W_i\|_2^2} \leq \max\left\{1, \sum_{i=1}^n \|W_i\|_F^2\right\} \left(\sum_{i=1}^n \frac{1}{\|W_i\|_2}\right)^2$$

by Cauchy-Schwarz inequality. The most left expression is the term that is in Neyshabur et al. [40] and the most right expression is the one we have in our bound.

However, the advantage of our result is that it does not depend on the assumption of all activation functions being equal to ReLU and that we can *compose* it with results for other networks to get bounds for compositions of networks.

**MPGNNs** As we discussed in the main text the difference between our bound and the one in Liao et al. [33] comes from the dependency on the weights.

Their dependency: $\max\left\{\zeta^{-(l+1)}, (\lambda\xi)^{\frac{l+1}{l}}\right\}$, where $\zeta = \min\{\|W_1\|_2, \|W_2\|_2, \|W_l\|_2\}$, $\lambda = \|W_1\|_2\|W_l\|_2, \xi = C_\phi \frac{(d\mathcal{C})^{l-1}-1}{d\mathcal{C}-1}$. Let us show, how our bound depends on $\lambda, \zeta$ and $\xi$:

$$\zeta^{-1} = \max\left\{\frac{1}{\|W_1\|_2}, \frac{1}{\|W_2\|_2}, \frac{1}{\|W_l\|_2}\right\} \geq \frac{1}{3}\left(\frac{1}{\|W_1\|_2} + \frac{1}{\|W_2\|_2} + \frac{1}{\|W_l\|_2}\right) \geq$$
$$\geq \frac{\min\{1, d\text{Lip}_\phi\text{Lip}_g\text{Lip}_\rho\}}{3}\left(\frac{1}{\|W_1\|_2} + \frac{1}{\min\{d\mathcal{C}, \|W_2\|_2\}} + \frac{1}{\|W_l\|_2}\right)$$

and

$$\zeta^{-1} = \max\left\{\frac{1}{\|W_1\|_2}, \frac{1}{\|W_2\|_2}, \frac{1}{\|W_l\|_2}\right\} \leq \frac{1}{\|W_1\|_2} + \frac{1}{\|W_2\|_2} + \frac{1}{\|W_l\|_2} \leq$$
$$\leq \frac{1}{\|W_1\|_2} + \frac{1}{\min\{d\mathcal{C}, \|W_2\|_2\}} + \frac{1}{\|W_l\|_2}.$$

Note that $\lambda\xi = T$. Now let us analyze two cases:

1. $(\lambda\xi)^{\frac{l+1}{l}} \geq (\zeta)^{-(l+1)}$. In this case $\zeta^{-1} \leq (\lambda\xi)^{1/l}$ and $\zeta^{-1}\lambda\xi \leq (\lambda\xi)^{\frac{l+1}{l}} = \max\{\zeta^{-(l+1)}, (\lambda\xi)^{\frac{l+1}{l}}\}$

2. $(\zeta)^{-(l+1)} \geq (\lambda\xi)^{\frac{l+1}{l}}$. In this case $\zeta^{-l} \geq \lambda\xi$ and $\zeta^{-1}\lambda\xi \leq \zeta^{-(l+1)} = \max\{\zeta^{-(l+1)}, (\lambda\xi)^{\frac{l+1}{l}}\}$

So, in the case when $d\text{Lip}_\phi\text{Lip}_g\text{Lip}_\rho$ is lower bounded by some constant we can conclude that asymptotically our bound is not inferior, than the one in Liao et al. [33].

## G Dependency on Model Parameters and Hyperparameters

In the Table 6 we present the dependency of existing results and our result on the model parameters and hyperparameters.

## H Learnable Filtration Functions

Fixed filtration functions dominate the PH/ML literature. The widespread use of learnable functions is a relatively recent phenomenon in PH-based ML, and usually runs orders of magnitude slower compared to non-learnable ones. Arguably, applying non-learnable functions still represents the mainstream approach in TDA.

Some works have explicitly advocated for fixed filtration functions (with learnable vectorizations) over learnable filtrations. Filtration functions can come in different flavors; for instance, they can rely on node degree [18], cliques [43], or node attributes [22]. Some of the popular options are parameter-free. Also, while some works showed gains using learnable filtrations [20], others have reported no benefits and adopted fixed functions instead [5, 34]. There is still no consensus about the significance of the gains associated with learnable filtration in many applications.

Perslay [5] uses fixed filtration functions. Despite the generality of our results, we provide specific bounds for PersLay, which employs fixed filtration functions.

Our work lays a strong foundation for analyzing learnable filtrations. One way to analyze PH with learnable filtration schemes could be to get upper bounds on perturbation of outputs in terms of the

Table 6: **The dependency of the bound on the model parameters.** All models have maximum width of weight matrices $h$. We consider $n$-layer multilayer perceptron (MLP) with weights $W_1, ..., W_n$. We consider $n$-layer GCN with weights $W_1, ..., W_n$. We consider $n$-layer ($n > 2$) MPGNN with weights $W_1, W_2, W_3$. We consider simple graphs with maximum degree $d$. We denote $\mathcal{C} = \text{Lip}_\phi \text{Lip}_g \text{Lip}_\rho \|W_2\|$, $\lambda = \|W_1\|_2 \|W_3\|_2$ and $\xi = ((d\mathcal{C})^{n-1}-1)/(d\mathcal{C}-1)$ and $|w|_2$ is the norm of all parameters in the model. Comparing to [33] we do not add $\text{Lip}_\phi$ to $\xi$ and instead of $W_l$ we have $W_3$. We consider PersLay with one of the classical point-transformation (described in [5]) and weight function with weights $W^{(\varphi)}, W^{(\omega)}$ and $\|\mathbf{w}\|_2$ is the norm of all parameters in the model. We consider abstract model that satisfy Lemma 2 requirements with $\mathbf{w}$, $T$, $\bar{\eta}$, $C_{norm}$ and $S_i$,

| Name (Reference) | Model Parameters | Model Hyperparameters |
|---|---|---|
| MLP, [40] | $\prod_{i=1}^{n} \|W_i\|_2 \sum_{i=1}^{n} \|W_i\|_F/\|W_i\|_2$ | $n\sqrt{h \ln(nh)}$ |
| GCN, [33] | $\prod_{i=1}^{n} \|W_i\|_2 \sum_{i=1}^{n} \|W_i\|_F/\|W_i\|_2$ | $n\sqrt{d^{n-1} h \ln(nh)}$ |
| MPGNN, [33] | $\|\mathbf{w}\|_2 \max\{\zeta^{-(n+1)}, (\lambda\xi)^{(n+1)/n}\}$ | $n\sqrt{h \ln(nh)}$ |
| PersLay | $\|\mathbf{w}\|_2 \|W^{(\varphi)}\|_2 \|W^{(\omega)}\|_2$ | $\sqrt{h^{3/2} \ln h}$ |
| Lemma 2 | $\|\mathbf{w}\|_2 T \sum_{i=1}^{n} 1/S_i$ | $C_{norm} \bar{\eta}^{-1} \sqrt{h \ln(nh)}$ |

filtration function parameters. This would additionally require an analysis of Wasserstein distances between persistence diagrams obtained with different parameters. We believe that for a specific class of graphs we can get modified upper bounds for perturbation with respect to filtration function parameters that would depend on Wasserstein distance of the same order. This additional analysis could be readily integrated into our framework to get generalization bounds for learnable filtrations.

# I  Discussion on the max over 1 and parameters norm.

From the first glance it may seem that introducing $\max\{1, \|w\|_2^2\}$ is suboptimal; however, $\|w\|_2^2$ is greater than 1 in many real-world scenarios. For instance, suppose we have parameters within the range of $\mathcal{O}(\varepsilon)$; then the squared norm would be greater than 1 if the number of parameters exceeds $\frac{1}{\varepsilon^2}$. Considering a typical choice in the Deep Learning literature, $\varepsilon = 0.01$, we need more than $10,000$ parameters. For example, a two-layer MLP with a hidden dimension of 64 (also a typical choice) has more than $8,000$ parameters, so in practice, $\|\mathbf{w}\|_2^2 \geq 1$ is usually true.

# J  Implementation details

## J.1  Datasets

Table 7 reports summary statistics of the datasets used in this paper.

Table 7: Statistics of the datasets.

| Dataset | #graphs | #classes | Avg #nodes | Avg #edges |
|---|---|---|---|---|
| NCI1 | 4110 | 2 | 29.87 | 32.30 |
| IMDB-B | 1000 | 2 | 19.77 | 96.53 |
| PROTEINS (full) | 1113 | 2 | 39.06 | 72.82 |
| MUTAG | 188 | 2 | 17.93 | 19.79 |
| DHRF | 756 | 2 | 42.43 | 44.54 |
| MOLHIV | 41127 | 2 | 25.5 | 27.5 |
| NCI109 | 4127 | 2 | 29.68 | 32.13 |

## J.2 Models

For the experiments with PersLay Classifier, we closely follow the filtration functions used in [5]. In particular, we use Kernel heat functions with parameters $t = 0.1$ and $t = 10$ for the remaining datasets. Instead of processing each diagram type using separate models, we combine ordinary and extended diagrams for 0- and 1-dimensional features and apply a single model. We use mean aggregation function in all experiments, and Gaussian point transformations. For the feedforward part of PersLay, we apply ReLU activation functions. All models are trained with Adam [27] and learning rate of $10^{-3}$ for 3000 epochs.

For the experiments with GNNs with persistence, we use graph isomorphism networks (GINs) with 2 layers and 64 hidden units. The models were trained for 2000 epochs using the Adam optimizer.

**Dependence on model paramaters.** Regarding the dependence on the spectral norm of weights, we reported results for a model with a final MLP (multilayer perceptron) of 2 hidden layers (3 layers in total) and width of 128 for all layers. The number of parameters of the point transformation was $h = 100$. For the experiments on width vs. generalization gap, we used $h = 100$, and 1 hidden layer with a varying width in $\{32, 64, 128, 256, 512\}$.

**Regularizing PersLay.** For the experiments regarding ERM and spectral norm regularizers, we perform model selection for (number of layers) $l \in \{2, 3\}$, and $\alpha_r \in \{10^{-3}, 10^{-4}, 10^{-5}, 10^{-6}\}$. Again, we use Gaussian point transformation, $h = 100$, and width equals to 128. Our goal was to see if we could observe gains from the regularized version even for shallow neural networks.

**Regularizing GNNs with persistence.** Here, we consider GNNs with 64 hidden units of 64 (width) and 2 layers. We set the dimensionality of PersLay parameters equal to 100, Gaussian point transformation, and mean aggregation function. We apply hold-out model selection with penalty term $\alpha_r \in \{10^{-5}, 10^{-6}, 10^{-7}, 10^{-8}\}$ using the validation set.

**Hardware.** For all experiments, we use Tesla V100 GPU cards and consider a memory budget of 32GB of RAM.

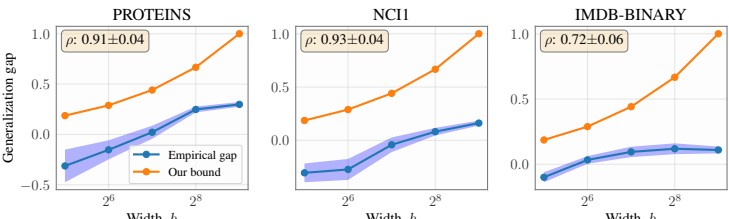

Figure 7: **Width vs. generalization gap for the triangle point transformation.**

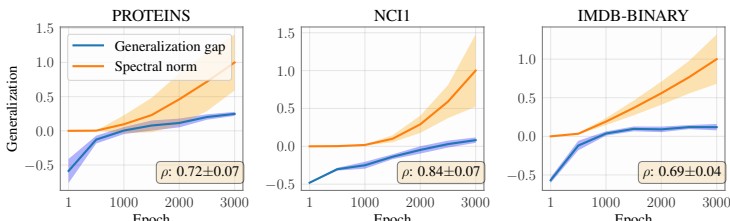

Figure 8: **Spectral norm vs. generalization gap for the triangle point transformation.**

## K  Additional visualizations

Figure 7 and Figure 8 report additional results for the triangle point transformation on the three largest datasets: PROTEINS, NCI1, and IMDB-BINARY. In particular, Figure 7 shows the dependence of the generalization on width, while Figure 8 shows the dependence on the spectral norm. Overall, our bound can capture the trend in the empirical gap and produces high correlation values for all datasets.

