# OpenReview forum: "Compositional PAC-Bayes: Generalization of GNNs with persistence and beyond"
_NeurIPS.cc/2024/Conference — NeurIPS 2024 poster_

### Official Review · Reviewer_ynbU · 2024-06-29

**Soundness:** 3
**Presentation:** 3
**Contribution:** 2
**Rating:** 4
**Confidence:** 3

**Summary:**

This paper derives novel generalization bounds for a special class of GNNs augmented with persistent homology descriptors, specifically PersLay. They empirically test the bound and use the bound to propose a regularization, which works well in practice.

**Strengths:**

1. The paper provides a clear background introduction to persistence homology and its integration with GNNs.
2. The derived bound has positive correlation with the empirical generalization gap of GNNs.
3. The paper is well positioned within related works on perturbation based generalization bounds of MLPs and GNNs.

**Weaknesses:**

1. It is hard to parse the significance of this work as the bound applies only to a restricted family of models augmented with PersLay. This issue could be mitigated by demonstrating that a model derived from this analysis is comparable with more advanced GNN models (not necessarily with PH), which the paper fails to do.
2. The paper would benefit from listing all assumptions used in the derivation.
3. The experiments could be strengthened by considering different GNN models augmented by different PersLay variants.

**Questions:**

How does the bound for GNNs with PH compare with their counterparts without PH? Does the result provide any insights into why PH is beneficial for enhancing GNN performance?

**Limitations:**

Please refer to the weaknesses section.

---

> ### Author Rebuttal · Authors · 2024-08-06
>
> Thanks for your feedback. We reply to your comments/questions below.
>
> > It is hard to parse the significance of this work as the bound applies only to a restricted family of models augmented with PersLay. This issue could be mitigated by demonstrating that a model derived from this analysis is comparable with more advanced GNN models (not necessarily with PH), which the paper fails to do.
>
> Thank you for the opportunity to clarify important aspects of our work.
>
> The main result (Lemma 2) and the Corollaries about model compositionality apply to a broad class of models and are not restricted to models augmented with PersLay. We used GNNs combined with PersLay to showcase our developed recipe since, despite their increasing popularity [1,2,3,4], their generalization behavior is heavily underexplored.
>
> Moreover, our analysis can be extended to other GNN architectures, such as GraphSAGE. Indeed, given the similarity between GraphSAGE and GCN, our analysis already subsumes GraphSAGE, as we can upper-bound GraphSAGE by leveraging full neighborhoods.
>
> We have included additional experiments with different PH-augmented GNNs (GCN, GraphSAGE, and GIN) in Table 2 of the rebuttal PDF. The results show the benefits of our bound used as a regularizer — the regularized variants achieve smaller generalization gaps and lower errors in most experiments.
>
> > The paper would benefit from listing all assumptions used in the derivation.
>
> Thank you for this comment. While our main result (Lemma 2) does not make any assumptions beyond data being iid, we agree that the perturbation analysis requires certain assumptions. We will gladly include all assumptions in the Appendix and overview them in the main text.
>
> The following list provides the assumptions:
> - Data (i.e. tuples) are i.i.d samples from some unknown distribution $\mathcal{D}$.
> - the width of all layers is bounded by $h$ (stated in Lemma 2).
> - For MLP: all inputs are contained in the $\ell_2$-ball of radius $B$.
> - For GCNs and MPGNNs: graphs are simple with maximum degree of $d$ and node features are contained in $\ell_2$-ball of radius $B$. (listed in the caption for Table 2 in the main paper)
> - For PersLay: the norm of the elements of persistence diagrams are contained in a $\ell_2$-ball with a radius $B$ and all of the considered point transformations and weight functions are Lipschitz continuous with respect to the parameters.
>
> > The experiments could be strengthened by considering different GNN models augmented by different PersLay variants.
>
> Thanks for your suggestion. We have run additional experiments using regularized versions of GraphSAGE, GIN, and GCN combined with PersLay (see Table 2 in the rebuttal PDF). Overall, our results show that the regularized methods achieve smaller generalization gaps and slightly lower classification errors. We will add these experiments to the revised manuscript.
>
> > How does the bound for GNNs with PH compare with their counterparts without PH? Does the result provide any insights into why PH is beneficial for enhancing GNN performance?
>
> From Informal Lemma 4 (two models in parallel), $C_{norm}$ and $C_{pert}$ (as well as $T$) of the combined bound scale as the maximum of the corresponding parameters of the GNN and PersLay bounds. Since GNNs usually have more parameters than PH in practice, the bound for the combined network would scale with the GNN bound, so PH does not introduce a lot of overhead in the generalization performance.
>
> We also note that combining PH with GNNs indeed improves the latter's expressivity [3,5] --- for instance, persistence diagrams contain information about the number of components and cycles that 1-WL (Weisfeiler-Leman) GNNs can not decide. However, while the expressivity of PH-augmented GNNs has been explored [5,6], their generalization capabilities remain largely uncharted, which is the motivation for our work.
>
> [1] PersLay. AISTATS 2020.
>
> [2] Topological neural networks go persistent, equivariant, and continuous. ICML, 2024.
>
> [3] Topological graph neural networks. ICLR 2022.
>
> [4] Position: Topological Deep Learning is the New Frontier for Relational Learning. ICML, 2024.
>
> [5] Going beyond persistent homology using persistent homology. NeurIPS 2023.
>
> [6] On the Expressivity of Persistent Homology in Graph Learning. arXiv, 2023
>
> ---
> We're grateful for your feedback. We hope our answers have addressed your concerns and improved your assessment of this work.

---

> > ### Author Response · Authors · 2024-08-14
> >
> > Dear reviewer,
> >
> > Thank you again for your constructive feedback.
> >
> > As detailed above, we've acted on all your comments and suggestions (including clarifying how our main result applied broadly significantly beyond PersLay, listing all assumptions, and providing results of additional experiments that demonstrate improved generalization with regularized PH-augmented GraphSAGE, GCN, GIN). We will include all these in the updated version. Please also see our global response, where we summarize the key steps taken to address the concerns of all the reviewers.
> >
> > We believe acting on your feedback has helped us consolidate our contributions, and reinforced the strengths of this work. Since only a few hours are left until the end of the discussion period,  we would greatly appreciate if you could update your score to reflect the same. Many thanks!
> >
> > Best regards.

---

### Official Review · Reviewer_Hv3h · 2024-07-11

**Soundness:** 3
**Presentation:** 3
**Contribution:** 3
**Rating:** 6
**Confidence:** 2

**Summary:**

The paper introduces a novel Compositional PAC-Bayes framework that addresses challenges related to the heterogeneity of Graph Neural Network (GNN) layers and persistent vectorization components. It provides data-dependent generalization bounds for PH vectorization schemes and persistence-augmented GNNs, offering insights into improved classifier design and generalization performance predictions.

**Strengths:**

1. The introduction of the Compositional PAC-Bayes framework is a significant contribution to the field, especially in handling heterogeneous GNN layers.
2.  The provision of data-dependent generalization bounds adds a valuable dimension to the analysis of GNNs and persistence-augmented models.
3.  The empirical evaluations on real-world datasets demonstrate the practical applicability and effectiveness of the proposed framework.

**Weaknesses:**

Clarity: Some sections of the paper may require further clarification to enhance readability and understanding for a broader audience. From experiments, we can observe that there exists a considerable gap between the theoretical results and empirical results.

**Questions:**

1. This paper mentioned the expressivity of PH and GNNs for many times. Does the expressivity have anything to do with the generalization?
2. What is the influence of using different filtration functions in the proposed framework?
3. Does the proposed framework have any concrete applications? Can you provide a case study?

**Limitations:**

yes

---

> ### Author Rebuttal · Authors · 2024-08-07
>
> Thank you for your feedback. We hope our answers below satisfactorily address your concerns. Otherwise, we will be happy to engage further.
>
> > Clarity: Some sections of the paper may require further clarification to enhance readability and understanding for a broader audience.
>
> Thanks for your comment. To further improve the clarity of our manuscript, we will:
> - include a list with the assumptions related to PAC-Bayes and the perturbation analysis adopted in the paper;
> - add a discussion about the main takeaways of our analysis, especially regarding how we can use it to choose better hyperparameters (see last answer to reviewer toK8).
>
> Should the reviewer provide additional suggestions for improvements, we would be happy to incorporate them.
>
> > From experiments, we can observe that there exists a considerable gap between the theoretical results and empirical results.
>
> Thanks for your comment. Overall, we note that our theoretical bounds strongly correlate with empirical generalization gaps for most datasets and models. This is what one would expect, given that theoretical generalization bounds are often loose.
> To complement our analysis, we have included additional experiments with different PH-augmented GNNs (GCN, GraphSAGE, and GIN) in Table 2 of the rebuttal PDF. The results reinforce that the regularized variants achieve smaller generalization gaps. We also report empirical vs. theoretical generalization plots for GraphSAGE — the results again confirm the practical relevance of our work.
>
> > This paper mentioned the expressivity of PH and GNNs for many times. Does the expressivity have anything to do with the generalization?
>
> Thank you for the opportunity to clarify this.
>
> In fact, expressivity and generalization can be at odds with each other, and finding a good tradeoff holds the key to success of machine learning models [1]. Indeed, enhancing expressivity typically comes at the expense of generalization. [2] established such a result for Graph Neural Networks, showing that the VC-dimension of GNNs with $L$ layers is lower bounded by the maximal number of graphs that can be distinguished by 1-WL (Weisfeiler-Leman test for isomorphism). High VC-dimension directly translates to poor generalization, whereas by definition greater the number of graphs that can be distinguished greater the expressivity. Thus, [2] showed the tension between expressivity and generalization in the context of message passing GNNs that are at most as expressive as the 1-WL test.
>
> Expressivity of machine learning models is certainly important [3]; however, it does not guarantee that powerful models that do well on training data would generalize, i.e., predict well on unseen data as well.
>
> Moreover, most of the generalization bounds in the literature share a common structure: population risk is bounded by empirical risk + complexity of the model class ($\approx$ expressivity). This fact also suggests the intricate connection between generalization and expressivity.
>
> > What is the influence of using different filtration functions in the proposed framework?
>
> Our work considers fixed filtration functions, and the exact choice of filtration function does not affect the bound. This happens because the considered filtration functions are parameter-free --- this part of the model does not impact the output change due to parameter perturbations. However, the choice of the filtration function affects the risk, not the bound on the difference.
>
> Also please see the global response for the discussion about learnable filtration functions.
>
> > Does the proposed framework have any concrete applications? Can you provide a case study?
>
> As a general result, Lemma 2 and the Corollaries about model compositionality of heterogeneous layers apply to a broad class of models. Indeed, in the paper, we show specific case studies where we can recover existing bounds (for MLPs, GCNs, and MPNNs) from our framework. In addition, we use our framework to derive new bounds for PH-augmented GNNs and PersLay --- as case studies.
>
> From an empirical standpoint, we leverage our results to derive integrated regularization procedures for different methods, including PersLay and PH-augmented GNNs. Our results show that the regularized variants can achieve better (test) classification performance and smaller empirical generalization gaps. In the rebuttal PDF (see Table 2), we provide additional results for different GNN architectures to further support our claims.
>
> [1] Generalization and Representational Limits of Graph Neural Networks. ICML, 2020
>
> [2] WL meet VC. arXiv, 2023
>
> [3] A Survey on The Expressive Power of Graph Neural Networks. arXiv , 2020
>
> Many thanks again for your thoughtful comments, which have helped us reinforce the strengths of this work.

---

> > ### Comment · Reviewer_Hv3h · 2024-08-08
> >
> > Thanks for all the authors' efforts to address my other concerns.  I have no further questions. I am positive to an acceptance.

---

> > > ### Author Response · Authors · 2024-08-08
> > >
> > > We are glad that our answers addressed your concerns and that you are positive about acceptance. Thank you again for your review and for acknowledging our rebuttal.

---

### Official Review · Reviewer_toK8 · 2024-07-12

**Soundness:** 3
**Presentation:** 4
**Contribution:** 3
**Rating:** 7
**Confidence:** 2

**Summary:**

This paper introduces a novel compositional PAC-Bayes framework for analyzing the generalization of heterogeneous machine learning models, with a particular focus on graph neural networks (GNNs) augmented with persistent homology (PH) features. The work develops a general PAC-Bayes lemma for heterogeneous models that not only recovers existing bounds for neural networks and GNNs but also extends them to more complex architectures. Notably, it provides the first data-dependent generalization bounds for PH-based models.
The key innovation lies in the compositional approach, presenting lemmas that allow for combining bounds from different model components. This enables the analysis of complex architectures like GNNs augmented with PH features, bridging an important gap in the theoretical understanding of topology-based graph representation learning methods. The framework's versatility is demonstrated by recovering existing bounds for various models and deriving new bounds for PersLay and its variants. The empirical evaluations across multiple datasets validate the theoretical results, showing correlation between the derived bounds and observed generalization performance.

**Strengths:**

- Novel theoretical contributions that advance the state-of-the-art in generalization analysis for GNNs and PH-based methods.
- Flexible framework that recovers existing bounds and enables analysis of new model compositions

**Weaknesses:**

- Empirical evaluation focuses mostly on graph classification tasks; additional experiments on node classification or link prediction could strengthen the results

**Questions:**

- The theoretical analysis assumes fixed filtration functions for PH. How limiting is this in practice, and could the framework be extended to learnable filtrations?

**Limitations:**

- While the experiments cover several datasets, they focus primarily on graph classification tasks. The paper could benefit from a broader range of experiments, including node classification or link prediction tasks, to demonstrate the generality of the approach.
- While the paper derives a regularization scheme from the bounds, it doesn't fully explore how the theoretical results could guide the design of better GNN+PH architectures in general. Some discussion on how the bounds suggest optimal ways to combine GNNs and PersLay could enhance the practical impact of the work.

These points did not diminish the overall contribution of the paper but addressing them could significantly strengthen its impact and applicability.

---

> ### Author Rebuttal · Authors · 2024-08-06
>
> Thank you for the feedback and for appreciating our work. We hope that the answers below sufficiently address your concerns. Otherwise, we would be happy to engage further.
>
> > Empirical evaluation focuses mostly on graph classification tasks; additional experiments on node classification or link prediction could strengthen the results
>
> Thanks for your comment. While we agree that developing bounds (and  running experiments) for different tasks would be valuable, our work focuses on graph-level prediction tasks. Extending it to node-level tasks, for instance, would require adapting the theoretical framework — the i.i.d. assumption (basic assumption in PAC-Bayes) may not hold. Since our experiments aim to validate and demonstrate the practical relevance of our analysis, we abide by the settings we consider in Sections 3 and 4. We also note that PH-augmented GNNs for node classification tasks often apply local topological descriptors [e.g., 1], which fundamentally differ from what we discuss elsewhere in the paper.
>
> > The theoretical analysis assumes fixed filtration functions for PH. How limiting is this in practice, and could the framework be extended to learnable filtrations?
>
> Thanks for your question. For an in-depth discussion about learnable filtrations, we kindly refer to the global response.
>
> > While the paper derives a regularization scheme from the bounds, it doesn't fully explore how the theoretical results could guide the design of better GNN+PH architectures in general. Some discussion on how the bounds suggest optimal ways to combine GNNs and PersLay could enhance the practical impact of the work.
>
> Indeed, this would strengthen our work; thank you for pointing this out! From our theoretical findings, we can say:
> - since the $C_{pert}$ constant for PersLay depends on the square root of its dimension, one should choose it significantly smaller than the dimension of the GNN to avoid an $O(h\sqrt{\ln h})$ dependency of the total generalization bound on $h$ instead of $O(\sqrt{h \ln h})$;
> - compared to the “k-Max” and “Mean” functions, the “Sum” aggregation function introduces a $\max\limits_{G\in\mathcal{G}} card(G)$ term to the bound, which in practice can be rather large. So, we recommend using “Mean” instead of “Sum”.
>
> Moreover, using our analysis, we can compare different PersLay variants (see Table 3 in the paper). Let us consider the constant weighting function for simplicity. Then, $C_{pert} = 2C_{norm}$, and as a result, we can rank different PersLay variants (e.g., k-landscapes, images, and silhouettes) by simply comparing their associated constants $C_{pert}$. For landscapes, $C_{pert} = 2\cdot 3\cdot B\sqrt{h}$; for images, we have $2 \cdot card \cdot \max \\{\sqrt{h}, \frac{1}{\tau e^{1/2}}\\}$; for silhouettes we have $2 \cdot card \max \\{B \sqrt{h}, \frac{1}{\tau e^{1/2}} \\}$. If the last two options use ‘sum’ as an aggregating function, then our generalization analysis suggests that k-landscapes would have stronger guarantees. If the last two options use ‘mean’ as an aggregating function, then $C_{pert}$ for k-landscapes would be at most $C_{pert}$ for silhouettes, and the result of comparison of k-landscapes and images could be in favor of both landscapes and images depending on chosen parameters $\tau$ and $B$.
>
> We will add this discussion to the revised manuscript.
>
> [1] Persistence Enhanced Graph Neural Network. AISTATS, 2020.
>
> We are grateful for your constructive feedback. Many thanks!

---

> > ### Comment · Reviewer_toK8 · 2024-08-14
> >
> > Thank you for the feedback and clarification. I will keep my score.

---

### Official Review · Reviewer_zp7n · 2024-07-14

**Soundness:** 3
**Presentation:** 4
**Contribution:** 3
**Rating:** 6
**Confidence:** 3

**Summary:**

This paper presents a compositional PAC learning framework for bounding the generalization gap in deep graph networks that are augmented by PersLay-vectorization of persistent homology features. Topological features can be complementary to deep features. Empirically, this combination can boost the empirical test performance. This paper investigates whether this observation reflects to the theory, and concludes that it does, up to some assumptions, for example in filtrations.

**Strengths:**

- To the best of my knowledge, this is the first generalization bound of its kind, presenting bounds for a heterogeneous network composed of PH vectorization and GNNs.
- I really like the presentation and exposition in this paper, that makes it informative and easy to follow.
- The idea of using a compositional pac-bayes framework for bounding generalization in PH-augmented GNNs really makes sense to me.
- Although I have not been able to thoroughly check all the proofs, the theoretical parts I investigated made sense.
- Given the nicheness of the topic, the models (GNN + PersLay) are intuitive for integrating topological features in deep graph learning. I have seen such architectures a few times.

**Weaknesses:**

- One obvious drawback is that the architectures of concern in this paper are rather limited and not the ones used in practice. While this limits the applicability, the paper's theoretical contributions make it less of a concern. According to me, speaking of generalization for PH-augmented GNNs through the lens of compositional pac-bayes is interesting in its own right.

- Most importantly, does the proposed bound, similar to the KL-terms in Neyshabur's bound, depend on the number of parameters in the network? (I was suspecting this due to the use of the norm.) If so, it is definitely worth discussing, as the recent theory of deep learning states that the intrinsic dimension matters rather than the ambient dimension. And if not, please clarify.

- I'm curious about the limitations of the compositional framework. Can we leverage the nature of PH and the integration to do something more specific to tighten the bound? The compositional framework seems to adapt a rather late fusion.

-  Ln. 81: The generalization error is defined to be the population risk and not as the difference between them. I believe some sort of a difference between test/train would be more appropriate. Why is this chosen? Is this common convention?

- I see PH as a hand-crafted way of extracting topological features, which goes a bit orthogonal to the current trends. Can the same analysis be extended to hybrid classical & topological deep learning (like the ones presented in [*,**]) which also operates on complexes? If so, do the authors see a straightforward way?

- There are multiple ways to combine PH(-vectorization) and GNNs. A two-branch architecture + PersLay is definitely a good way. Yet, I would expect that the paper investigates other forms of combination. For example, what about learning a filtration?

- I'm noticing that only a single family of GNNs is used in the paper. Is it possible to show results with more modern architectures at least like GraphSage?

- There is also a rather recent literature of leveraging PH to bound generalization [***] (other way around), which this paper does not seem to discuss. It would be good to have both sides of this picture to stress on the impact of PH in the generalization theory.

[*] Hajij, Mustafa et al. "Topological deep learning: Going beyond graph data." arXiv preprint arXiv:2206.00606 (2022).

[**] Papamarkou, Theodore, Tolga Birdal, Michael M. Bronstein, Gunnar E. Carlsson, Justin Curry, Yue Gao, Mustafa Hajij et al. "Position: Topological Deep Learning is the New Frontier for Relational Learning." In Forty-first International Conference on Machine Learning. 2024.

[***] Birdal, Tolga, Aaron Lou, Leonidas J. Guibas, and Umut Simsekli. "Intrinsic dimension, persistent homology and generalization in neural networks." Advances in Neural Information Processing Systems 34 (2021): 6776-6789.

**Questions:**

I have appended my questions after each relevant weakness. I would be happy if the authors could address them. In addition:
- Why would the bounds worsen with the increase number of epochs (where I believe training gets better)?
- There is also growth in the width disproportionately to the empirical gap. Why would this happen?

In general, some more light on the empirical findings would be useful.

**Limitations:**

The paper appropriately discusses the limitations and remains to be largely theoretical in a very niche domain of machine learning. As such, I don't see any issues with broader impact.

---

> ### Author Rebuttal · Authors · 2024-08-06
>
> Many thanks for your thoughtful, constructive, and insightful review. We hope that the answers below sufficiently address your concerns.
>
> > the architectures of concern in this paper are rather limited and not the ones used in practice.
>
> Despite the generality of our framework, we focused on PH and PH-augmented GNNs due to the lack of generalization bounds for these classes of models and their increasing popularity in the graph learning community. In this regard, a prominent way to combine persistence diagrams (PDs) with GNNs consists of leveraging PDs as global topological descriptors, which are then concatenated (in parallel) with graph-level GNN embeddings (see [1]). We followed this approach in our work.
>
> Importantly, our work paves the way for the generalization analysis of different classes of topological neural networks and their integration with PH [2,3]. For instance, compositional PAC-Bayes may be an asset in analyzing models that exploit 0-dim PD as node-level information that can be combined with node embeddings at each GNN layer (see [4]).
>
> > does the proposed bound [...] depend on the number of parameters in the network?
>
> Like Neyshabur’s bound, our results implicitly depend on the number of model parameters via model hyper-parameters (e.g., number of layers) and parameter values (e.g., spectral norms). We show in Table 1 of the rebuttal PDF the dependence of our bounds on model parameters and hyperparameters separately.
>
> While we initially found PAC-Bayes particularly suitable to develop a general recipe for composing bounds for heterogeneous layers, we agree that applying our ideas to other generalization frameworks (e.g., in terms of intrinsic dimension) can provide further insights into generalization in DL. We believe this is a fascinating research direction for future work.
>
> > Can we leverage the nature of PH and the integration to do something more specific to tighten the bound?
>
> In this work, we focused more on deriving a flexible recipe that can accommodate a broad class of models and less on tightening bounds. However, extending our ideas regarding the compositionality of heterogeneous layers to other generalization paradigms (e.g., PH-dim [6]) is a very interesting direction, and it seems a promising approach to get tighter bounds.
>
> > The generalization error is defined to be the population risk and not as the difference between them.
>
> Indeed, it is possible to define it as a difference, but we wanted to be consistent with some reference works in the PAC-Bayesian literature, such as [7,8,9].
>
> > Can the same analysis be extended to hybrid classical & topological DL … which also operates on complexes?
>
> Indeed, we believe that Lemma 2 can be used to derive bounds for higher-order TNNs [2, 5] and their combination with the classical PH approach. Intuitively, we would expect T (a relevant component in Lemma 2) to depend on the norms of the weights associated with the different neighborhood structures. We note that the technical details to ensure that the conditions in Lemma 2 are satisfied need to be figured out. We believe this is an interesting future work.
>
> > What about learning a filtration?
>
> For a discussion about learnable filtration functions, please see the global response.
>
> > Is it possible to show results with more modern architectures at least like GraphSage?
>
> We have run additional experiments using GraphSage (see Tab 2 in the rebuttal PDF). Although our bound is not particularly tailored to GraphSAGE, due to its similarity to GCNs (GraphSAGE samples neighbors at every iteration instead of counting on all neighbors), our regularization scheme also benefits GraphSAGE. Table 2 also contains additional results regarding regularized versions of GCN and GIN combined with PersLay. Overall, the regularized methods achieve smaller generalization gaps and lower errors in most experiments.
>
> From a theoretical perspective, we can upper-bound GraphSAGE by leveraging full neighborhoods — obtaining GCNs. In some sense, our analysis already subsumes GraphSAGE. In fact, our additional experiments using GraphSAGE show that the empirical generalization gap strongly correlates with our bound (see Fig 2 in the rebuttal PDF). However, achieving tighter bounds would require deriving a specific perturbation analysis for GraphSAGE.
>
> > There is also a rather recent literature of leveraging PH to bound generalization [..] which this paper does not seem to discuss.
>
> We agree that using PH to bound generalization is an important line of work. Indeed, Appendix G of our paper discusses it. We will also appropriately position these influential works in the main text in the revised manuscript.
>
> > Why would the bounds worsen with the increase number of epochs?
>
> The bounds worsen because the spectral norms increase during training to fit the training data. To validate this, we now report the average spectral norms for GNN and MLP layers in Fig 1 of rebuttal PDF.
>
> > There is also growth in the width disproportionately to the empirical gap. Why would this happen?
>
> We suspect that some dependencies in our bounds could be improved, which may explain the discrepancy you noted.
>
> [1] Going beyond persistent homology using persistent homology. NeurIPS 2023.
>
> [2] Topological deep learning: Going beyond graph data. Arxiv, 2022.
>
> [3] Topological neural networks go persistent, equivariant, and continuous. ICML, 2024.
>
> [4] Topological graph neural networks. ICLR 2022.
>
> [5] Weisfeiler and Lehman Go Cellular: CW Networks. NeurIPS 2021.
>
> [6] Intrinsic Dimension, Persistent Homology and Generalization in Neural Networks, NeurIPS 2021.
>
> [7] A PAC-Bayesian Approach To Spectrally-Normalized Margin Bounds for Neural Networks. ICLR 2018
>
> [8] Simplified PAC-Bayesian Margin Bounds. Learning Theory and Kernel Machines, Lecture Notes in Computer Science, 2003
>
> [9] A PAC-Bayesian Approach to Generalization Bounds for Graph Neural Networks. ICLR 2021
>
> We're grateful for your constructive feedback. Many thanks!

---

> > ### Comment · Reviewer_zp7n · 2024-08-12
> >
> > I thank the authors for the good work and their explanations. I will maintain my recommendation of acceptance. Note that, in the newly provided plots, the gap between the empirical error and the proposed bound grows with epochs. This might be seen as a little concerning and stresses the importance of tightening the bounds in future works, for example through some of the directions I suggested.

---

### Author Rebuttal · Authors · 2024-08-07

We are grateful to all the reviewers for their time and insightful comments, as well as to the (senior) area, program, and general chairs for their service to the community.

We are pleased to note that reviewers appreciate the **novelty** (zp7n, toK8, Hv3h, ynbU) and the **presentation** (zp7n, toK8) of our work. Also reviewers found that our work provides a **flexible framework** (toK8) and **empirical evaluations on real-world datasets that demonstrate the practical applicability and effectiveness** (Hv3h).

To the best of our efforts, we have tried to address all the specific comments that have been raised by each reviewer.

Below, we provide some of the main revisions:

1. Reviewer zp7n asked about the dependency of our bounds on the number of model parameters. We used this opportunity to **clarify how PAC-Bayesian bounds depend on parameter values and hyper-parameters**. **Table 1** in the rebuttal PDF outlines these dependencies.

2. Reviewers zp7n and ynbU asked about other GNN architectures, such as GraphSAGE (zp7n). **We have run additional experiments considering GCN, GraphSAGE, and GIN combined with PersLay to assess the effectiveness of our theoretical bounds as a regularization scheme** (see **Table 2** of the rebuttal PDF). We considered 3 datasets (NCI1, NCI109, and PROTEINS) and reported test classification errors and empirical generalization gaps. Our results show that the regularized versions achieve competitive classification errors and significantly smaller generalization gaps. In addition, in Figure 2 (attached PDF), we show that our theoretical bound strongly correlates with the empirical gap for GraphSAGE.

3. Reviewer zp7n asked why the generalization bound increases with the number of epochs. To explain that, we report in **Figure 1 (attached PDF) the average spectral norms over training**. Interestingly, we observe that MLP layers dominate GNN ones — i.e., the average norm of MLPs is higher than that of GNNs.

4. Reviewer ynbU asked for clarification regarding the assumptions of our analysis. To summarize, **our main results (Lemma 2 and corollaries about compositionality) assume i.i.d data, while the remaining (model specific) results make typical assumptions related to perturbation analysis** (e.g., inputs lie in a $\ell_2$-norm ball, graphs have bounded degree). In the revised manuscript, we will list the assumptions for every result in the Appendix and overview them in the main text.

5. Reviewer toK8 suggested providing a discussion on takeaways from our theoretical analysis. **We will add the discussion about the choices of PersLay hyperparameters one can make informed by our theoretical bound**.

6.  Reviewers asked about analyzing learnable filtration functions. Below, we summarize the reasons behind our modeling choice as well as insights into how to extend results for the learnable filtration case:

    - **Fixed filtration functions dominate the PH/ML literature**. The widespread use of learnable functions is a relatively recent phenomenon in PH-based ML, and usually runs orders of magnitude slower compared to non-learnable ones. Arguably, applying non-learnable functions still represents the mainstream approach in TDA.

    - **Some works have explicitly advocated for fixed filtration functions (with learnable vectorizations) over learnable filtrations**. Filtration functions can come in different flavors; for instance, they can rely on node degree [1], cliques [2], or node attributes [3]. Some of the popular options are parameter-free. Also, while some works showed gains using learnable filtrations [4], others have reported no benefits and adopted fixed functions instead [5,6]. There is still no consensus about the significance of the gains associated with learnable filtration in many applications.

    - **Perslay [5] uses fixed filtration functions**. Despite the generality of our results, we provide specific bounds for PersLay, which employs fixed filtration functions.

    - **Our work lays a strong foundation for analyzing learnable filtrations**. One way to analyze PH with learnable filtration schemes could be to get upper bounds on perturbation of outputs in terms of the filtration function parameters. This would additionally require an analysis of Wasserstein distances between persistence diagrams obtained with different parameters. We believe that for a specific class of graphs we can get modified upper bounds for perturbation with respect to filtration function parameters that would depend on Wasserstein distance of the same order. This additional analysis could be readily integrated into our framework to get generalization bounds for learnable filtrations.

[1] Deep learning with topological signatures. NeurIPS 2017.

[2] Networks and cycles: A persistent homology approach to complex networks. ECCS 2013.

[3] Going beyond persistent homology using persistent homology. NeurIPS 2023.

[4] Topological GNNs. ICLR 2022.

[5] PersLay. AISTATS 2020.

[6] Improving Self-supervised Molecular Representation Learning using Persistent Homology. NeurIPS 2023.

---

We thank reviewers again for their very constructive comments.

---

### Comment · Area_Chair_FSUo · 2024-08-10
**to reviewers: please respond to rebuttal**

Dear reviewer,

Thank you for your reviews. The authors have tried to address your concerns in the rebuttal. If you have not already done so, please carefully read their rebuttal, and let them know what you think, and whether there is any more clarifications you require. Note that author-reviewer discussion ends on August 13th.
Thanks!
the AC

---

### Decision · Program_Chairs · 2024-09-25

**Decision:**

Accept (poster)

**Comment:**

This paper uses a PAC-Bayes framework to recover known generalization bounds for MLPs and GNNs, and derive new generalization bounds for GNNs with additional topological features. Experiments show that these bounds correlate well with empirical generalisation performance, and a regularization technique based on the analysis in this paper was shown to yield good results.

The reviewers have voiced some concerns regarding the nichenss of the particular topological GNNs used. Nonetheless, most reviewers found the technical contribution of generalization using PAC-Bayes important, and recommended acceptance.